PREPARED FOR SUBMISSION TO JHEP

# Exact multi-instantons in topological string theory

**Jie Gu**[a] **and Marcos Mariño**[b]

[a]*School of Physics and Shing-Tung Yau Center*
*Southeast University, Nanjing 210096, China*

[b]*Département de Physique Théorique et Section de Mathématiques*
*Université de Genève, Genève, CH-1211 Switzerland*

*E-mail:* eij.ug.phys@gmail.com, Marcos.Marino@unige.ch

ABSTRACT: Topological string theory has multi-instanton sectors which lead to non-perturbative effects in the string coupling constant and control the large order behavior of the perturbative genus expansion. As proposed by Couso, Edelstein, Schiappa and Vonk, these sectors can be described by a trans-series extension of the BCOV holomorphic anomaly equations. In this paper we find exact, closed form solutions for these multi-instanton trans-series in the case of local Calabi–Yau manifolds. The resulting multi-instanton amplitudes turn out to be very similar to the eigenvalue tunneling amplitudes of matrix models. Their form suggests that the flat coordinates of the Calabi–Yau manifold are naturally quantized in units of the string coupling constant, as postulated in large $N$ dualities. Based on our results we obtain a general picture for the resurgent structure of the topological string in the local case, which we illustrate with explicit calculations for local $\mathbb{P}^2$.

## 1  Introduction

String theory, as many other quantum theories, is characterized by factorially divergent perturbative expansions, and this suggests that the full theory should have instanton sectors which lead to exponentially small corrections in the string coupling constant $g_s$ [1, 2].

One class of string theories where instanton sectors have been relatively well understood is non-critical string theories. In some cases, the exact solution to non-critical strings is described by a non-linear ODE of the Painlevé type. The perturbative genus expansion corresponds to the conventional asymptotic expansion of the ODE near an irregular singular point, and multi-instanton sectors are given by the so-called trans-series solution to the ODE, i.e. a solution involving exponentially small corrections (see e.g. [2–5] for introductions to trans-series solutions). In addition, one can interpret these non-perturbative effects in terms of D-branes [6] and check in many situations that D-brane calculations reproduce important ingredients of the trans-series solution (see e.g. [7, 8] for examples of these calculations).

Another class of string theories where one can hope to understand in detail the structure of space-time instanton corrections is topological string theory with a Calabi–Yau (CY) target manifold. It was proposed in [9–11] that this can be achieved in the framework of the theory of resurgence, in particular by using the large order behavior of the perturbative series as a concrete, quantitative guide for the study of non-perturbative effects. A crucial step in this program was made by Couso, Edelstein, Schiappa and Vonk (CESV) in [12–14]. Topological string amplitudes satisfy a set of partial differential equations known as the holomorphic anomaly equations (HAE), or BCOV equations [15, 16]. The HAE give, among other things, a recursive procedure to calculate the perturbative genus expansion of topological string theory which can

be implemented in a very efficient way, specially in the local case [17, 18]. CESV noted in [12] that the BCOV equations could be regarded as the topological string analogue of the Painlevé equations in non-critical string theory, and they suggested that the multi-instanton amplitudes of the topological string can be obtained by considering trans-series solutions to the HAE. In [13] they managed to obtain explicit trans-series solutions in the case of local $\mathbb{P}^2$, the simplest non-trivial toric CY manifold. They also tested their results in detail against the large order behavior of the perturbative genus expansion. In addition, it was found in [19] that the instanton amplitudes of [13] provide the correct non-perturbative effects to match the non-perturbative definition of topological strings proposed in [20–22]. Therefore, there is little doubt that the trans-series found in [12, 13] are indeed the multi-instanton amplitudes of the topological string.

In spite of these successes, the actual solutions found by CESV have a number of drawbacks. The expressions they find are obtained recursively, order by order in the string coupling constant and in the instanton number, and their complexity grows very fast as one goes to higher order in $g_s$ or to higher instanton number. In addition, the physical content of the solution is obscure, as it is given by complicated expressions involving the BCOV propagators and the instanton actions, as well as their derivatives. This has made the theory of CESV difficult to work with, even for experts in the field.

In this paper we consider general local CY manifolds with one modulus, and we find *exact* multi-instanton solutions to the trans-series extension of the BCOV equations considered by CESV. Our solutions give explicit, simple expressions at all orders in the string coupling constant and for general instanton number, and they cover in particular all the examples studied in [13]. This is achieved by exploiting an operator formulation of the BCOV equations, which was proposed in [23, 24] to study a similar problem in the NS topological string [25]. In the holomorphic limit, these explicit solutions take a surprisingly simple and suggestive form: they can be written in terms of the perturbative free energy $\mathcal{F}(t)$ and its derivatives, and they involve an exponent of the form

$$\exp\left(\mathcal{F}(t - n\alpha g_s) - \mathcal{F}(t)\right). \tag{1.1}$$

This structure is typical of multi-instantons obtained by eigenvalue tunneling in matrix models [26–28] (here, $n$ is the instanton number, and $\alpha$ is a constant characterizing the instanton sector, to be defined more precisely below). It has also appeared in a related context in [29]. The expression (1.1) suggests that $t$, the flat coordinate of the CY, is quantized in units of $\alpha g_s$. In topological string theories with large $N$ duals this is an ingredient of the duality [22, 30, 31], but here it is deduced only from the holomorphic anomaly equations.

The focus of this paper is on the formal structure of the trans-series solutions associated to the possible Borel singularities. This is a preliminary ingredient in order to understand the full resurgent structure of topological string theory, which requires additional ingredients. First of all, we have to determine which of the possible Borel singularities are actually realized, and we have to know the corresponding alien derivatives (or, equivalently, the values of the Stokes constants for the actual singularities). In particular, one would like to know how this structure changes as we vary the moduli. In this paper we address this second set of problems (in an incomplete way) in the example of local $\mathbb{P}^2$, mostly to illustrate our results. By using long perturbative series for the topological string free energies we explore the structure of Borel singularities, and we use our exact multi-instanton solutions to calculate some alien derivatives.

In a companion paper we use similar techniques and ideas to study the resurgent structure of quantum periods [32].

This paper is organized as follows. In section 2 we review the HAE and its rôle in generating

the perturbative series of topological string theory. In section 3 we present the CESV framework and we proceed to construct our exact solutions, first in the one-instanton case and then in the general multi-(anti)-instanton case. We end up with a conjecture on the structure of alien derivatives and how they relate to our exact solutions. In section 4 we illustrate some of our considerations with explicit examples in the case of the local $\mathbb{P}^2$ manifold. Some details on topological string theory on this toric CY can be found in Appendix A. In the final section we present our conclusions and some open problems.

## 2   Topological strings and the holomorphic anomaly equations

In this paper we will consider topological string theory with a target given by a toric Calabi–Yau manifold $X$ (for introductions to topological string theory, see e.g. [33, 34]). For simplicity, we will consider the case in which one has a single true modulus (although one can have many "mass parameters," see e.g. [35]). The basic observables are the genus $g$ topological string free energies $\mathcal{F}_g(t)$, where $t$ is a flat coordinate parametrizing the one-dimensional moduli space. These free energies represent contributions of genus $g$ Riemann surfaces to the free energy of string theory. It is then natural to consider the total free energy, given by the formal power series

$$\mathcal{F}(t, g_s) = \sum_{g \geq 0} \mathcal{F}_g(t) g_s^{2g-2}. \tag{2.1}$$

Although we have not indicated it explicitly, these free energies depend on a choice of electro-magnetic duality frame. There is in principle an infinity of choices, related by $SL(2, \mathbb{Z})$ transformations. At the level of free energies, changes of frame are implemented by generalized Fourier transforms [36]. There are also canonical choices of frame, associated to special points or regions in moduli space. Two important points are the so-called *large radius point* and the *conifold point*. Near the large radius point, the appropriate flat coordinate goes to infinity $t \to \infty$, and one has the expansion

$$\mathcal{F}_g(t) = p_g(t) + \sum_{d \geq 1} N_{g,d} \, \mathrm{e}^{-dt}, \tag{2.2}$$

where $N_{g,d}$ are Gromov–Witten invariants, and $p_g(t)$ are polynomials in $t$ (of degree 3 for $g = 0$, degree 1 for $g = 1$, and degree 0 for $g \geq 2$). In the large radius frame the free energies are completely captured by enumerative geometry.

In the conifold frame, the appropriate flat coordinate is denoted by $t_c$, and it is chosen so as to vanish at the conifold point. Near this point, and in the conifold frame, the free energies for $g \geq 2$ have the following structure,

$$\mathcal{F}_g^c(t_c) = \mathfrak{a} \, \mathfrak{b}^{g-1} \frac{B_{2g}}{2g(2g-2)} t_c^{2-2g} + \mathcal{O}(1), \tag{2.3}$$

where $B_{2g}$ are Bernoulli numbers, and $\mathfrak{a}$, $\mathfrak{b}$ are constants depending on the model. The behaviour of the free energies at the conifold locus is a fundamental ingredient of the theory and it has been argued to be universal [37]. It will play an important rôle in what follows.

The topological string free energies can be upgraded to more general, non-holomorphic functions of the moduli of the CY $X$, as emphasized in the seminal papers [15, 16]. We will denote the non-holomorphic free energies as $F_g$, in order to distinguish them from their holomorphic versions $\mathcal{F}_g$. In the case of $g = 0$, $F_0$ is itself holomorphic, so $\mathcal{F}_0 = F_0$ and we will use both notations interchangeably. The general formalism to analyze the non-holomorphic free energies

is based on the special geometry of the Calabi–Yau moduli space and it was developed in [16] (see e.g. [33, 34] for reviews). Here we will consider a simplified version of the formalism which is appropriate for a one-modulus local CY manifold.

In their non-holomorphic version, the free energies are regarded as functions of a complex coordinate $z$, which parametrizes the moduli space, and of a propagator function $S$, which encodes all the non-holomorphic dependence. The free energies will then be denoted by $F_g(S, z)$, $g \geq 2$. We also note that $z$ is related to the flat coordinate by a mirror map $t(z)$. The genus one free energy is slightly different, and in fact it defines the propagator $S$ through the equation

$$\partial_z F_1 = \frac{1}{2} C_z S + \text{holomorphic}. \tag{2.4}$$

Here, $C_z$ denotes the so-called Yukawa coupling in the $z$ coordinate, which is defined by

$$\partial_t^3 F_0 = C_t = \left(\frac{\mathrm{d}z}{\mathrm{d}t}\right)^3 C_z. \tag{2.5}$$

The holomorphic part in the r.h.s. of (2.4) can be regarded as a choice of "gauge" for the propagator. We will choose it to vanish, so that we simply have

$$S = \frac{2}{C_z} \partial_z F_1. \tag{2.6}$$

The holomorphic limit of the free energies $F_g(S, z)$ is obtained by taking the holomorphic limit of the propagator, which will be denoted by $\mathcal{S}$. It is a function of $z$. We then have

$$\mathcal{F}_g(t) = F_g\left(S = \mathcal{S}(z), z\right), \tag{2.7}$$

after one expresses $z$ as a function of $t$ through the inverse mirror map. One powerful aspect of this formalism is that the propagator $S$ has different holomorphic limits depending on the frame, so that the holomorphic free energies in a given frame can be obtained from the *same* function $F_g(S, z)$ by choosing different holomorphic limits for $S$ and different inverse mirror maps.

As we mentioned before, a choice of frame can be specified by an appropriate choice of a flat coordinate $t$. There is a very useful formula which expresses the holomorphic limit of $\mathcal{S}$ in terms of the mirror map $t(z)$ for the corresponding flat coordinate:

$$\mathcal{S} = -\frac{1}{C_z} \frac{\mathrm{d}^2 t}{\mathrm{d}z^2} \frac{\mathrm{d}z}{\mathrm{d}t} - \mathfrak{s}(z). \tag{2.8}$$

Here, $\mathfrak{s}(z)$ is a holomorphic function of $z$ which is independent of the frame. Due to (2.6), one also obtains a formula for the holomorphic limit of $F_1$,

$$\mathcal{F}_1 = -\frac{1}{2} \log\left(\frac{\mathrm{d}t}{\mathrm{d}z}\right) + f_1(z). \tag{2.9}$$

Here, $f_1(z)$ is a holomorphic function which is related to the function $\mathfrak{s}(z)$ appearing in (2.8) by

$$\mathfrak{s}(z) = -\frac{1}{C_z} \tilde{f}(z) = -\frac{2}{C_z} \frac{\mathrm{d}f_1(z)}{\mathrm{d}z}. \tag{2.10}$$

The propagator satisfies various important properties which will be much used. The first one, which follows from the special geometry of the moduli space, is that its derivative w.r.t. $z$ can be written as a quadratic polynomial in $S$

$$\partial_z S = S^{(2)}, \qquad S^{(2)} = C_z\left(S^2 + 2\mathfrak{s}(z)S + \mathfrak{f}(z)\right), \tag{2.11}$$

where $\mathfrak{f}(z)$ is again a universal, holomorphic function independent of the frame. Another property of the propagator which will be needed is the following. The CY moduli space is a special Kähler manifold, and in particular it has a Levi–Civita connection associated to the Kähler metric. In the one-dimensional case, the corresponding Christoffel symbol turns out to be related to the propagator by the equation

$$\Gamma^z_{zz} = -C_z \left( S + \mathfrak{s}(z) \right). \tag{2.12}$$

Finally, we will need the following property. Let us denote by $\mathcal{S}_1, \mathcal{S}_2$ the holomorphic limits of the propagator in frames associated to the flat coordinates $t_1$, $t_2$, respectively. We first note that we can always write $t_2$ as a linear combination of periods in the first frame

$$t_2 = \alpha \, \partial_{t_1} F_0(t_1) + \beta t_1 + \gamma, \tag{2.13}$$

where $\alpha$, $\beta$, $\gamma$ are constants. Then, a simple calculation shows that the difference between the holomorphic propagators is

$$\mathcal{S}_1 - \mathcal{S}_2 = \alpha \left( \frac{\mathrm{d}t_1}{\mathrm{d}z} \frac{\mathrm{d}t_2}{\mathrm{d}z} \right)^{-1}. \tag{2.14}$$

Let us now write down the holomorphic anomaly equations of BCOV, in the case at hand. These equations determine the dependence of $F_g(S, z)$ on the propagator, once the lower order functions $F_{g'}(S, z)$, $g' < g$, are known. In order to use the HAE properly, it is important to note that the derivative w.r.t. $z$ of $S$ can be traded for a polynomial in $S$, as we have seen in (2.11). For this reason, it is useful to introduce an operator $\mathfrak{D}_z$. This is a derivation, and it acts as follows on a function of $S$ and $z$:

$$\mathfrak{D}_z f(S, z) = \partial_S f(S, z) S^{(2)} + \partial_z f(S, z), \tag{2.15}$$

where $S^{(2)}$ was defined in (2.11). Note that, in the holomorphic limit, $S$ becomes a function of $z$, $\mathcal{S}(z)$, and $\mathfrak{D}_z$ becomes $\partial_z$. An important property of $\mathfrak{D}_z$ is that it does not commute with $\partial_S$, and we rather have

$$[\partial_S, \mathfrak{D}_z] = \left( \partial_S S^{(2)} \right) \partial_S = 2C_z(S + \mathfrak{s})\partial_S. \tag{2.16}$$

The HAE read then,

$$\frac{\partial F_g}{\partial S} = \frac{1}{2} \left( \mathcal{D}_z \mathfrak{D}_z F_{g-1} + \sum_{m=1}^{g-1} \mathfrak{D}_z F_m \mathfrak{D}_z F_{g-m} \right), \qquad g \geq 2. \tag{2.17}$$

Here, $\mathcal{D}_z$ is the covariant derivative w.r.t. the Kähler metric, and it acts on the indexed object $\mathfrak{D}_z F_g$ as

$$\mathcal{D}_z \mathfrak{D}_z F_g = \mathfrak{D}_z^2 F_g - \Gamma^z_{zz} \mathfrak{D}_z F_g. \tag{2.18}$$

It is straightforward to calculate e.g. $F_2(S, z)$ from this equation. The starting point is (2.6), and we immediately find

$$F_2(S, z) = C_z^2 \left\{ \frac{5S^3}{24} + \frac{S^2}{24} \left( 9\mathfrak{s}(z) + 3\frac{C_z'}{C_z^2} \right) + \frac{1}{4} S \mathfrak{f}(z) \right\} + f_2(z), \tag{2.19}$$

where $f_2(z)$ is an arbitrary holomorphic function of $z$ which is not fixed by (2.17), and it appears as an integration constant. This happens at all genera: the HAE determine $F_g(S, z)$ up to an arbitrary holomorphic function $f_g(z)$ called the *holomorphic ambiguity*. Fixing this ambiguity is

the main obstacle to solve the topological string in the BCOV approach, and additional information is required. In the case of local CY manifolds with one modulus, it was shown in [18] that this can be done by using the behavior (2.3) at the conifold point. This makes it possible to calculate the free energies $F_g$ to high order in the genus.

It is convenient to reformulate the HAE in terms of a "master equation" for the full perturbative series. Various equations of this type have been used in the literature [12, 16, 38], but here we will use a simpler version similar to what was proposed in [23]. We first introduce the modified free energy

$$\widetilde{F} = g_s^2 F - F_0 = \sum_{g \geq 1} F_g g_s^{2g}, \tag{2.20}$$

as well as

$$\widehat{F} = \widetilde{F} - g_s^2 F_1 = \sum_{g \geq 2} F_g g_s^{2g}. \tag{2.21}$$

The master equation is then,

$$\frac{\partial \widehat{F}}{\partial S} = \frac{g_s^2}{2} \mathcal{D}_z \mathfrak{D}_z \widetilde{F} + \frac{1}{2} \left( \mathfrak{D}_z \widetilde{F} \right)^2. \tag{2.22}$$

## 3 Multi-instanton amplitudes

### 3.1 Trans-series extension of the holomorphic anomaly equations

The discussion in the previous section was restricted to the perturbative sector of the theory. However, we expect the full topological string to contain spacetime instantons, perhaps originating in topological D-branes, and leading to exponentially small corrections in the string coupling constant. Let us consider the perturbative free energies in a given frame, characterized by the flat coordinate $t$. Then, the $\ell$-th instanton amplitude is expected to be of the form

$$\mathcal{F}^{(\ell)}(t) \approx \exp \left( -\ell \frac{\mathcal{A}}{g_s} \right), \qquad \ell \in \mathbb{Z}_{>0}. \tag{3.1}$$

Here, $\mathcal{A}$ is the instanton action, and general considerations suggest that $\mathcal{A}$ must be a combination of CY periods [12, 13, 39]. We will write it as

$$\mathcal{A} = \alpha \partial_t \mathcal{F}_0(t) + \beta t + \gamma. \tag{3.2}$$

More precisely, we expect the action $\mathcal{A}$ to belong to a lattice of periods, therefore the coefficients $\alpha$, $\beta$ and $\gamma$ should satisfy some integrality properties. In addition, the instanton action can be physically interpreted as the mass of a D-brane, since 1, $t$, and $\partial_t \mathcal{F}_0$ are the masses of D0, D2 and D4 branes (in an appropriate normalization). However, these considerations will not be needed in what follows. More important to our purposes is the fact that $\mathcal{A}$ can be interpreted as a flat coordinate and therefore it defines a frame. We will often denote the corresponding propagator as $\mathcal{S}_\mathcal{A}$, which can be written as [13]

$$\mathcal{S}_\mathcal{A} = -\frac{1}{C_z} \frac{\partial_z^2 \mathcal{A}}{\partial_z \mathcal{A}} - \mathfrak{s}(z). \tag{3.3}$$

Let us also note that, if $\alpha = 0$, the frame defined by $\mathcal{A}$ is the same frame in which one calculates the free energies, since $\mathcal{A}$ coincides with $t$ up to a shift and a rescaling.

The calculation of multi-instanton amplitudes (3.1) is challenging, since it is not even clear what is the framework to use. Here we will follow the successful strategy of [12, 13], which is inspired by the theory of ODEs of Écalle. The starting point of [12, 13] is the holomorphic anomaly equation for the non-holomorphic perturbative free energies, in the form of a master equation like (2.22). In order to solve this equation, one uses, instead of a perturbative ansatz, a general *trans-series ansatz*, i.e. an ansatz of the form,

$$F = \sum_{\ell \geq 0} \mathcal{C}^\ell F^{(\ell)}. \tag{3.4}$$

Here,

$$F^{(0)} = \sum_{g \geq 0} F_g(S, z) g_s^{2g-2} \tag{3.5}$$

is the perturbative free energy, $\mathcal{C}$ is a trans-series parameter which keeps track of the exponential order, and $F^{(\ell)}$ are non-holomorphic versions of the multi-instanton amplitudes. We will assume that they have the structure

$$F^{(\ell)} = \mathrm{e}^{-\ell \mathcal{A}/g_s} \sum_{k \geq 0} g_s^k F_k^{(\ell)}, \tag{3.6}$$

where $F_k^{(\ell)}$ depends on $S$, $z$ and $\mathcal{A}$.

It is instructive to work out the very first orders in $g_s$ of the first instanton correction, following [12, 13]. To do this, we plug the ansatz (3.4) in the master equation (2.22). We note from (2.20) and (2.21) that

$$\widehat{F}^{(\ell)} = \widetilde{F}^{(\ell)} = g_s^2 F^{(\ell)}, \qquad \ell \geq 1, \tag{3.7}$$

since the subtraction of the genus 0, 1 terms does not have any effect in the instanton amplitudes. It is easy to see that the first instanton correction has to satisfy the linear equation

$$\frac{\partial F^{(1)}}{\partial S} = \frac{g_s^2}{2} \mathcal{D}_z \mathfrak{D}_z F^{(1)} + \mathfrak{D}_z \widetilde{F}^{(0)} \mathfrak{D}_z F^{(1)}. \tag{3.8}$$

As noted in [12, 13], the first consequence of these equations is that

$$\partial_S \mathcal{A} = 0, \tag{3.9}$$

which is expected since $\mathcal{A}$ is a holomorphic period and it only depends on $z$. We find the following recursive equations for the coefficients of $F^{(1)}$:

$$\partial_S F_n^{(1)} = \frac{1}{2} \left( \mathfrak{D}_z \mathcal{A} \right)^2 F_n^{(1)} + \frac{1}{2} \left( \mathfrak{D}_z^2 - \Gamma_{zz}^z \mathfrak{D}_z \right) F_{n-2}^{(1)} - \frac{\mathfrak{D}_z \mathcal{A}}{2} \left( C_z(S - \mathcal{S}_\mathcal{A}) + 2\mathfrak{D}_z \right) F_{n-1}^{(1)}$$
$$- \mathfrak{D}_z \mathcal{A} \sum_{\ell=1}^{\left[\frac{n+1}{2}\right]} \mathfrak{D}_z F_\ell^{(0)} F_{n+1-2\ell}^{(1)} + \sum_{\ell=1}^{\left[\frac{n}{2}\right]} \mathfrak{D}_z F_\ell^{(0)} \mathfrak{D}_z F_{n-2\ell}^{(1)}, \tag{3.10}$$

where $\mathcal{S}_\mathcal{A}$ was introduced in (3.3) (note that, since $\mathcal{A}$ does not depend on $S$, $\mathfrak{D}_z \mathcal{A} = \partial_z \mathcal{A}$). Explicitly, we have, for $n = 0, 1$,

$$\partial_S F_0^{(1)} = \frac{1}{2} \left( \mathfrak{D}_z \mathcal{A} \right)^2 F_0^{(1)},$$
$$\partial_S F_1^{(1)} = \frac{1}{2} \left( \mathfrak{D}_z \mathcal{A} \right)^2 F_1^{(1)} - \frac{\mathfrak{D}_z \mathcal{A}}{2} \left( C_z(S - \mathcal{S}_\mathcal{A}) + 2\mathfrak{D}_z \right) F_0^{(1)} - \mathfrak{D}_z \mathcal{A} \mathfrak{D}_z F_1^{(0)} F_0^{(1)}. \tag{3.11}$$

These equations were integrated in [12, 13], although only the first one has a simple solution

$$F_0^{(1)} = f_0^{(1)}(z) \exp\left(\frac{1}{2}\left(\mathfrak{D}_z \mathcal{A}\right)^2 S\right). \tag{3.12}$$

Here, $f_0^{(1)}(z)$ is an integration constant which can be regarded as the manifestation of the holomorphic ambiguity in the multi-instanton solutions. As in the perturbative case, we need additional conditions that fix this ambiguity. This was also addressed in [12, 13] in an ingenious way, which we now explain.

As in the perturbative case, one can evaluate the holomorphic limit of the multi-instanton solution in an arbitrary frame by simply setting $S$ to the appropriate value. It was pointed out in [12, 13, 19] that multi-instanton amplitudes associated to an instanton action $\mathcal{A}$ simplify in the frame defined by $\mathcal{A}$ itself, i.e. when $S = S_{\mathcal{A}}$. Let us consider for example the one-instanton case, and let us denote by $\mathcal{F}_{\mathcal{A}}^{(1)}$ the holomorphic limit in that frame. Then, one has

$$\mathcal{F}_{\mathcal{A}}^{(1)} = (\mathcal{A} + g_s)\, e^{-\mathcal{A}/g_s}, \tag{3.13}$$

up to an overall multiplicative constant which can be absorbed in the trans-series parameter. The behavior (3.13) can be justified in the case in which $\mathcal{A}$ is proportional to the conifold flat coordinate $t_c$ and we work in the conifold frame. As is well-known in the theory of resurgence, the one-instanton amplitude should govern the large order behavior of the perturbative series. If $t_c$ is sufficiently small, we expect the large order behavior of $\mathcal{F}_g^c$ to be dominated by the pole of order $2g - 2$ in (2.3). By using the well-known formula for the Bernoulli numbers,

$$B_{2g} = (-1)^{g-1} \frac{2(2g)!}{(2\pi)^{2g}} \sum_{\ell \geq 1} \ell^{-2g}, \tag{3.14}$$

one finds the following all-orders asymptotic behavior for the polar part in (2.3):

$$\frac{\mathfrak{a}}{2\pi^2} \Gamma(2g-1) \sum_{\ell \geq 1} (\ell \mathcal{A}_c)^{1-2g}\left(\mu_{0,\ell} + \frac{\ell \mathcal{A}_c}{2g-2}\mu_{1,\ell}\right). \tag{3.15}$$

Here,

$$\mathcal{A}_c = \frac{2\pi i}{\sqrt{\mathfrak{b}}} t_c, \qquad \mu_{0,\ell} = \frac{\mathcal{A}_c}{\ell}, \qquad \mu_{1,\ell} = \frac{1}{\ell^2}. \tag{3.16}$$

By using the standard correspondence between large order behavior and exponentially small corrections (see e.g. [40]), one sees that (3.15) corresponds to an $\ell$-th instanton amplitude of the form,

$$(\mu_{0,\ell} + g_s \mu_{1,\ell})\, e^{-\ell \mathcal{A}_c/g_s}, \tag{3.17}$$

up to overall factors which do not depend on $\ell$. For $\ell = 1$ we obtain (3.13). In addition, this suggests a generalization of the boundary condition (3.13) to the $\ell$-th instanton case,

$$\mathcal{F}_{\mathcal{A}}^{(\ell)} = \left(\frac{\mathcal{A}}{\ell} + \frac{g_s}{\ell^2}\right) e^{-\ell \mathcal{A}/g_s}. \tag{3.18}$$

Although the above argument applies only to the conifold frame, somewhat surprisingly the result can be checked to be true as well for the large radius frame [19]. The "trivial" $\ell$-th instanton

amplitude (3.18) first appeared in [41], in the study of the resurgent structure of topological string theory on the resolved conifold.

Let us now come back to (3.12). If we now use the boundary condition (3.13) we can easily fix the value of $f_0^{(1)}$, and the final result is

$$F_0^{(1)} = \mathcal{A} \exp\left(\frac{1}{2}\left(\mathfrak{D}_z\mathcal{A}\right)^2\left(S - \mathcal{S}_\mathcal{A}\right)\right).\tag{3.19}$$

The next term might be obtained with some additional effort, and one finds

$$
\begin{aligned}
F_1^{(1)} = -\,&\mathcal{A}\mathrm{e}^{\frac{1}{2}(\mathfrak{D}_z\mathcal{A})^2(S-\mathcal{S}_\mathcal{A})}\Big\{\frac{1}{6}C_z\left(\mathfrak{D}_z\mathcal{A}\left(S - \mathcal{S}_\mathcal{A}\right)\right)^3 + \frac{1}{2}C_z\mathfrak{D}_z\mathcal{A}\left(S - \mathcal{S}_\mathcal{A}\right)^2 \\
&\qquad\qquad\qquad\qquad + \frac{1}{2}C_z\mathfrak{D}_z\mathcal{A}\,\mathcal{S}_\mathcal{A}(S - \mathcal{S}_\mathcal{A})\Big\} \\
+\,&\mathrm{e}^{\frac{1}{2}(\mathfrak{D}_z\mathcal{A})^2(S-\mathcal{S}_\mathcal{A})}\left\{1 - (\mathfrak{D}_z\mathcal{A})^2\left(S - \mathcal{S}_\mathcal{A}\right)\right\}.
\end{aligned}\tag{3.20}
$$

Higher order terms become more and more complicated. In spite of this complexity, it was checked very carefully in [13] that the above one-instanton amplitude describes correctly the large order behavior of the perturbative free energies (even away from the holomorphic limit).

Multi-instanton amplitudes can be also computed with the same method. The HAE equation for the $\ell$-th instanton amplitude is

$$\frac{\partial F^{(\ell)}}{\partial S} = \frac{g_s^2}{2}\mathcal{D}_z\mathfrak{D}_zF^{(\ell)} + \mathfrak{D}_z\widetilde{F}^{(0)}\mathfrak{D}_zF^{(\ell)} + \frac{g_s^2}{2}\sum_{r=1}^{\ell-1}\mathfrak{D}_zF^{(r)}\mathfrak{D}_zF^{(\ell-r)},\tag{3.21}$$

where we have taken into account (3.7). However, as noted in [13], in the multi-instanton case it becomes more and more difficult to calculate the higher order corrections in $g_s$. In this paper we will obtain exact, closed formulae for $F^{(\ell)}$ to all orders in $g_s$. The physics of the instanton amplitudes will be also much more transparent in our expressions.

### 3.2 Operator formulation

The key idea to obtain our exact solutions is to reformulate the HAE in terms of a pair of differential operators which were introduced in [23, 24], in the context of the NS topological string.

The first operator is defined as

$$\mathsf{D} = T\,\mathfrak{D}_z\tag{3.22}$$

where

$$T = \mathfrak{D}_z\mathcal{A}\left(S - \mathcal{S}_\mathcal{A}\right).\tag{3.23}$$

To understand the meaning of this operator, let us evaluate it in the holomorphic limit, in the frame whose flat coordinate is $t$. Let us assume that the action is given by (3.2). Then, by using the relation between propagators (2.14), we find

$$T \to \alpha\frac{\mathrm{d}z}{\mathrm{d}t},\tag{3.24}$$

so that

$$\mathsf{D} \to \alpha\partial_t.\tag{3.25}$$

Finally, another important fact is that $T$ vanishes in the frame $S = \mathcal{S}_{\mathcal{A}}$, and therefore the action of $\mathsf{D}$ also vanishes in that frame.

The second operator is

$$\mathsf{W} = T^2 \mathcal{D}_S, \qquad \mathcal{D}_S = \partial_S - \mathfrak{D}_z \widetilde{F}^{(0)} \mathfrak{D}_z. \tag{3.26}$$

We note that $\mathsf{D}$, $\mathcal{D}_S$ and $\mathsf{W}$ are all derivations. We also introduce the following object:

$$\mathcal{G} = \mathcal{A} + \mathsf{D}\widetilde{F}^{(0)}. \tag{3.27}$$

This has again an appealing interpretation. First of all, we point out that, when $\alpha \neq 0$, the instanton action $\mathcal{A}$ defines a modified prepotential:

$$\mathcal{F}_0^{\mathcal{A}}(t) = \mathcal{F}_0(t) + \frac{\beta}{2\alpha}t^2 + \frac{\gamma}{\alpha}t, \tag{3.28}$$

in such a way that

$$\mathcal{A} = \alpha \partial_t \mathcal{F}_0^{\mathcal{A}}. \tag{3.29}$$

(3.28) involves a redefinition of the prepotential by a quadratic term in the flat coordinate, which is allowed since it does not change the Yukawa coupling. When considering multi-instanton amplitudes, it will be convenient to redefine the prepotential as prescribed by (3.28), and we will omit the superscript $\mathcal{A}$. With this redefinition, we find that

$$\mathcal{G} \rightarrow \alpha g_s^2 \partial_t \mathcal{F}(t, g_s) \tag{3.30}$$

in the holomorphic limit. The formal series in the r.h.s. can be regarded as a "quantum period," i.e. as a $g_s$ deformation of the classical period $\partial_t \mathcal{F}_0$ (this should not be confused with the quantum periods appearing in the NS limit of the topological string).

The most important property of the two operators $\mathsf{W}$, $\mathsf{D}$ is the commutation relation

$$[\mathsf{W}, \mathsf{D}] = \mathsf{D}\mathcal{G}\,\mathsf{D}, \tag{3.31}$$

which was already used in [23], in the context of the NS topological string. We will now prove (3.31) for general local CY with one modulus. We first note the following identity

$$\frac{1}{2}(S - \mathcal{S}_{\mathcal{A}})\partial_S S^{(2)} \mathfrak{D}_z \mathcal{A} - (S^{(2)} - \mathfrak{D}_z \mathcal{S}_{\mathcal{A}})\mathfrak{D}_z \mathcal{A} - (S - \mathcal{S}_{\mathcal{A}})\mathfrak{D}_z^2 \mathcal{A} = 0. \tag{3.32}$$

We also have the useful equalities,

$$\begin{aligned}
\mathfrak{D}_z T &= \mathfrak{D}_z^2 \mathcal{A}(S - \mathcal{S}_{\mathcal{A}}) + \mathfrak{D}_z \mathcal{A}(S^{(2)} - \mathfrak{D}_z \mathcal{S}_{\mathcal{A}}) = \frac{1}{2}\partial_S S^{(2)} T = -\Gamma_{zz}^z T, \\
\mathcal{D}_S T &= \mathfrak{D}_z \mathcal{A} - \mathfrak{D}_z \widetilde{F}^{(0)} \left( \mathfrak{D}_z^2 \mathcal{A}(S - \mathcal{S}_{\mathcal{A}}) + \mathfrak{D}_z \mathcal{A}(S^{(2)} - \mathfrak{D}_z \mathcal{S}_{\mathcal{A}}) \right).
\end{aligned} \tag{3.33}$$

Let $f$ be an arbitrary function of $S$ and $z$. A direct calculation gives

$$\mathcal{D}_S \mathsf{D}f = T\left(\mathfrak{D}_z + \partial_S S^{(2)}\right)\mathcal{D}_S f + \mathfrak{D}_z \mathcal{G} \mathfrak{D}_z f. \tag{3.34}$$

We then find

$$\mathsf{W}\,\mathsf{D}f = \mathsf{D}\mathcal{G}\,\mathsf{D}f + T^3\left(\mathfrak{D}_z + \partial_S S^{(2)}\right)\mathcal{D}_S f. \tag{3.35}$$

At the same time,

$$\mathsf{D}\,\mathsf{W}f = T^3 \mathfrak{D}_z \mathcal{D}_S f + 2T^2 (\mathfrak{D}_z T) \mathcal{D}_S f. \tag{3.36}$$

By using (3.33) and (3.32), (3.31) follows. Another consequence of (3.33) is the crucial happy fact that

$$\mathsf{D}^2 = T^2 \left( \mathfrak{D}_z - \Gamma^z_{zz} \right) \mathfrak{D}_z = T^2 \mathcal{D}_z \mathfrak{D}_z, \tag{3.37}$$

and one reconstructs the covariant derivative by acting twice with $\mathsf{D}$.

It is now possible to write the original HAE and its trans-series extension by using only the operators $\mathsf{W}$ and $\mathsf{D}$. The master equation (2.22) reads, in this language,

$$\mathsf{W}\,\widehat{F}^{(0)} = \frac{g_s^2}{2}\mathsf{D}^2 \widetilde{F}^{(0)} - \frac{1}{2}\left(\mathsf{D}\widetilde{F}^{(0)}\right)^2 + g_s^2 \mathsf{D}F_1^{(0)}\mathsf{D}\widetilde{F}^{(0)}, \tag{3.38}$$

while the equation (3.21) for the $\ell$-th instanton amplitude $F^{(\ell)}$ becomes,

$$\mathsf{W}F^{(\ell)} = \frac{g_s^2}{2}\mathsf{D}^2 F^{(\ell)} + \frac{g_s^2}{2}\sum_{r=1}^{\ell-1}\mathsf{D}F^{(r)}\mathsf{D}F^{(\ell-r)}. \tag{3.39}$$

### 3.3  The one-instanton amplitude

The simpler instanton amplitude is of course $F^{(1)}$. It follows from (3.39) that it satisfies

$$\mathsf{W}F^{(1)} = \frac{g_s^2}{2}\mathsf{D}^2 F^{(1)}. \tag{3.40}$$

We will now prove that the exact solution to this equation, at all orders in $g_s$, with the correct boundary condition (3.13) is

$$F^{(1)} = \left(\mathcal{G} - g_s \mathsf{D}\Phi + g_s\right) \mathrm{e}^{-\Phi/g_s}, \tag{3.41}$$

where $\Phi$ is the formal power series

$$\Phi = \sum_{k\geq 1} \frac{(-1)^{k-1}g_s^{k-1}}{k!}\mathsf{D}^{k-1}\mathcal{G}. \tag{3.42}$$

We will present our proof in several steps. Our first claim is that

$$E = \mathrm{e}^{-\Phi/g_s} \tag{3.43}$$

satisfies (3.40). Equivalently, $\Phi$ satisfies the equation

$$\mathsf{W}\Phi = \frac{g_s^2}{2}\mathsf{D}^2\Phi - \frac{g_s}{2}\left(\mathsf{D}\Phi\right)^2, \tag{3.44}$$

To prove this, we have to establish first an intermediate result, namely

$$\mathsf{W}\mathcal{G} = \frac{g_s^2}{2}\mathsf{D}^2\mathcal{G}. \tag{3.45}$$

This is done by direct calculation. We have

$$\mathsf{W}\mathcal{G} = \mathsf{W}\mathcal{A} + g_s^2\mathsf{W}\mathsf{D}F_1^{(0)} + \mathsf{W}\mathsf{D}\widehat{F}^{(0)}. \tag{3.46}$$

We can now use the commutation relation (3.31) and (3.38) to write

$$\mathsf{W}\mathcal{G} = \mathsf{W}\mathcal{A} + g_s^2 \mathsf{W}\mathsf{D}F_1^{(0)} - g_s^2 \mathsf{D}\mathcal{G}\mathsf{D}F_1^{(0)} + \mathsf{D}\widetilde{F}^{(0)}\mathsf{D}\mathcal{A} + \frac{g_s^2}{2}\mathsf{D}^3\widetilde{F}^{(0)} + g_s^2\mathsf{D}\left(\mathsf{D}F_1^{(0)}\mathsf{D}\widetilde{F}^{(0)}\right). \quad (3.47)$$

On the other hand,

$$\frac{g_s^2}{2}\mathsf{D}^2\mathcal{G} = \frac{g_s^2}{2}\left(\mathsf{D}^2\mathcal{A} + \mathsf{D}^3\widetilde{F}^{(0)}\right). \quad (3.48)$$

By using the definition of $\mathcal{G}$ in the r.h.s. of (3.47) we conclude that

$$\mathsf{W}\mathcal{G} - \frac{g_s^2}{2}\mathsf{D}^2\mathcal{G} = g_s^2\left[-\frac{1}{2}\mathsf{D}^2\mathcal{A} + T^2\partial_S\left(\mathsf{D}F_1^{(0)}\right) - \mathsf{D}\mathcal{A}\mathsf{D}F_1^{(0)}\right]. \quad (3.49)$$

A direct calculation shows that the sum of the terms inside the bracket in the r.h.s. is zero. To see this, one uses (2.6) to write

$$\mathsf{D}F_1^{(0)} = \frac{1}{2}\mathfrak{D}_z\mathcal{A}C_z(S - \mathcal{S}_A)S, \quad (3.50)$$

and takes into account that

$$-\frac{1}{2}\mathsf{D}^2\mathcal{A} = -\frac{1}{2}(\mathfrak{D}_z\mathcal{A})^3(S - \mathcal{S}_A)^3 C_z, \qquad -\mathsf{D}\mathcal{A}\mathsf{D}F_1^{(0)} = -\frac{1}{2}(\mathfrak{D}_z\mathcal{A})^3 S(S - \mathcal{S}_A)^2 C_z. \quad (3.51)$$

We are now ready to prove that $\Phi$, as defined in (3.42), satisfies (3.44). To do this, we need to write $\Phi$ in the convenient form

$$\Phi = \mathsf{O}\mathcal{G}, \quad (3.52)$$

where $\mathsf{O}$ is the operator

$$\mathsf{O} = \sum_{k \geq 1} \frac{(-1)^{k-1}g_s^{k-1}}{k!}\mathsf{D}^{k-1} = \frac{1}{g_s\mathsf{D}}\left(1 - \mathrm{e}^{-g_s\mathsf{D}}\right) = \frac{1}{g_s}\int_0^{g_s}\mathrm{e}^{-u\mathsf{D}}\mathrm{d}u. \quad (3.53)$$

It is now clear that, in order to verify (3.44), we have to calculate the commutator of $\mathsf{W}$ with $\mathsf{O}$. We first calculate the commutator of $\mathsf{W}$ with $\mathrm{e}^{-u\mathsf{D}}$. This can be done with Hadamard's lemma,

$$\mathrm{e}^A B \mathrm{e}^{-A} = \sum_{n \geq 0} \frac{1}{n!}[A, B]_n, \quad (3.54)$$

where the iterated commutator $[A, B]_n$ is defined by

$$[A, B]_{n \geq 1} = [A, [A, B]_{n-1}], \qquad [A, B]_0 = B. \quad (3.55)$$

In our case, we have the simple result that

$$[\mathsf{D}, \mathsf{W}]_{n \geq 1} = -(\mathsf{D}^n\mathcal{G})\,\mathsf{D}, \quad [\mathsf{D}, \mathsf{W}]_0 = \mathsf{W}, \quad (3.56)$$

therefore

$$\mathsf{W}\,\mathrm{e}^{-u\mathsf{D}} = \mathrm{e}^{-u\mathsf{D}}\left(\mathsf{W} - \sum_{k \geq 1}\frac{u^k}{k!}\left(\mathsf{D}^k\mathcal{G}\right)\mathsf{D}\right). \quad (3.57)$$

The parentheses emphasize that $\mathsf{D}^k$ acts only on $\mathcal{G}$. By using now the integral formula for $\mathsf{O}$ we conclude that

$$\mathsf{W}\mathsf{O} = \mathsf{O}\mathsf{W} - \frac{1}{g_s} \int_0^{g_s} \mathrm{d}u \, \mathrm{e}^{-u\mathsf{D}} \left[ (\mathrm{e}^{u\mathsf{D}} - 1)\mathcal{G} \right] \mathsf{D}. \tag{3.58}$$

Note that

$$\mathsf{O}\mathsf{W}\mathcal{G} = \frac{g_s^2}{2} \mathsf{O}\mathsf{D}^2 \mathcal{G} = \frac{g_s^2}{2} \mathsf{D}^2 \Phi, \tag{3.59}$$

since $\mathsf{O}$ commutes with $\mathsf{D}^2$. We have to calculate now

$$\frac{1}{g_s} \int_0^{g_s} \mathrm{e}^{-u\mathsf{D}} \left\{ \left[ (\mathrm{e}^{u\mathsf{D}} - 1)\mathcal{G} \right] \mathsf{D}\mathcal{G} \right\} \mathrm{d}u. \tag{3.60}$$

Since $\mathsf{D}$ is a derivation, we have

$$\mathrm{e}^{-u\mathsf{D}}(fg) = \left( \mathrm{e}^{-u\mathsf{D}} f \right) \left( \mathrm{e}^{-u\mathsf{D}} g \right), \tag{3.61}$$

therefore

$$\mathrm{e}^{-u\mathsf{D}} \left\{ \left[ (\mathrm{e}^{u\mathsf{D}} - 1)\mathcal{G} \right] \mathsf{D}\mathcal{G} \right\} = \left[ (1 - \mathrm{e}^{-u\mathsf{D}})\mathcal{G} \right] \left[ \mathrm{e}^{-u\mathsf{D}} \mathsf{D}\mathcal{G} \right]. \tag{3.62}$$

On the other hand, we have that

$$\mathsf{D}\Phi = \sum_{k \geq 1} \frac{(-1)^{k-1} g_s^{k-1}}{k!} \mathsf{D}^k \mathcal{G} = \frac{1}{g_s} \int_0^{g_s} \mathrm{e}^{-u\mathsf{D}} \mathsf{D}\mathcal{G} \mathrm{d}u. \tag{3.63}$$

Its square can be computed as

$$(\mathsf{D}\Phi)^2 = \frac{1}{g_s^2} \int_0^{g_s} \mathrm{e}^{-u\mathsf{D}} \mathsf{D}\mathcal{G} \, \mathrm{d}u \int_0^{g_s} \mathrm{e}^{-v\mathsf{D}} \mathsf{D}\mathcal{G} \, \mathrm{d}v = \frac{2}{g_s^2} \int_0^{g_s} \mathrm{e}^{-u\mathsf{D}} \mathsf{D}\mathcal{G} \, \mathrm{d}u \int_0^u \mathrm{e}^{-v\mathsf{D}} \mathsf{D}\mathcal{G} \mathrm{d}v$$
$$= \frac{2}{g_s^2} \int_0^{g_s} \left[ \mathrm{e}^{-u\mathsf{D}} \mathsf{D}\mathcal{G} \right] \left[ (1 - \mathrm{e}^{-u\mathsf{D}})\mathcal{G} \right] \mathrm{d}u, \tag{3.64}$$

where, in going to the last line, instead of integrating the symmetric function in $v$, $u$ over the square $[0, g_s]^2$, we integrated it over the triangle below the diagonal, and multiplied the result by two. We then find, by combining (3.58), (3.59) and (3.64),

$$\mathsf{W}\mathsf{O}\mathcal{G} = \frac{g_s^2}{2} \mathsf{D}^2 \Phi - \frac{g_s}{2} (\mathsf{D}\Phi)^2, \tag{3.65}$$

which proves (3.44).

One could think that $E$, defined in (3.43), is the one-instanton amplitude we are looking for. However, although it satisfies the trans-series HAE, as we have seen, it does not satisfy the boundary condition (3.13), since

$$E_{\mathcal{A}} = \mathrm{e}^{-\mathcal{A}/g_s}. \tag{3.66}$$

Let us then consider an ansatz for $F^{(1)}$ of the form

$$F^{(1)} = \mathfrak{a} \, \mathrm{e}^{-\Phi/g_s}, \tag{3.67}$$

Then, $\mathfrak{a}$ has to satisfy the equation

$$\mathsf{W}\mathfrak{a} = \frac{g_s^2}{2} \mathsf{D}^2 \mathfrak{a} - g_s \mathsf{D}\mathfrak{a} \, \mathsf{D}\Phi \tag{3.68}$$

as well as the boundary condition

$$\mathfrak{a}_{\mathcal{A}} = \mathcal{A} + g_s. \tag{3.69}$$

We now claim that

$$\mathfrak{a} = \mathcal{G} - g_s\mathsf{D}\Phi + g_s \tag{3.70}$$

is the sought-for solution. To see it, we calculate

$$\mathsf{W}\mathfrak{a} = \mathsf{W}\mathcal{G} - g_s\mathsf{D}\mathsf{W}\Phi - g_s\mathsf{D}\mathcal{G}\mathsf{D}\Phi. \tag{3.71}$$

Acting with $\mathsf{D}$ on the equation satisfied by $\Phi$, we find

$$\mathsf{D}\mathsf{W}\Phi = \frac{g_s^2}{2}\mathsf{D}^3\Phi - g_s\mathsf{D}\Phi\mathsf{D}^2\Phi. \tag{3.72}$$

We also have

$$\mathsf{D}\mathfrak{a} = \mathsf{D}\mathcal{G} - g_s\mathsf{D}^2\Phi, \qquad \mathsf{D}^2\mathfrak{a} = \mathsf{D}^2\mathcal{G} - g_s\mathsf{D}^3\Phi. \tag{3.73}$$

Therefore,

$$\mathsf{W}\mathfrak{a} + g_s\mathsf{D}\mathfrak{a}\mathsf{D}\Phi - \frac{g_s^2}{2}\mathsf{D}^2\mathfrak{a} = \mathsf{W}\mathcal{G} - \frac{g_s^2}{2}\mathsf{D}^2\mathcal{G} = 0, \tag{3.74}$$

and (3.68) holds. We finally arrive at (3.41). It is an easy exercise to check explicitly that this reproduces the first two terms in the expansion that we obtained before, (3.19) and (3.20).

We can now evaluate (3.41) in the holomorphic limit associated to the flat coordinate $t$. In this limit, $\mathsf{D}$ becomes $\alpha\partial_t$, and

$$\frac{1}{g_s}\Phi \to \mathcal{F}(t) - \mathcal{F}(t - \alpha g_s), \qquad \mathfrak{a} \to g_s + \alpha g_s^2\left(\partial_t\mathcal{F}\right)(t - \alpha g_s), \tag{3.75}$$

so that

$$\mathcal{F}^{(1)} = \left(g_s + \alpha g_s^2\left(\partial_t\mathcal{F}\right)(t - \alpha g_s)\right)\exp\left\{\mathcal{F}\left(t - \alpha g_s\right) - \mathcal{F}(t)\right\}. \tag{3.76}$$

It is useful to write explicitly the very first terms of its expansion in powers of $g_s$:

$$\begin{aligned}
\mathcal{F}^{(1)} = \mathrm{e}^{-\mathcal{A}/g_s}\exp\left(\frac{\alpha^2}{2}\partial_t^2\mathcal{F}_0\right) \\
\times\left\{\mathcal{A} + g_s\left(1 - \alpha^2\partial_t^2\mathcal{F}_0 - \mathcal{A}\left(\alpha\partial_t\mathcal{F}_1 + \frac{\alpha^3}{6}\partial_t^3\mathcal{F}_0\right)\right) + \mathcal{O}(g_s^2)\right\}.
\end{aligned} \tag{3.77}$$

They give the holomorphic limit of the expressions (3.19) and (3.20). One important property of (3.76) is that it can be uniquely written in terms of the conventional, perturbative topological string free energy. There was some speculation that the instanton amplitudes obtained in [12, 13] could contain new geometric information [42], but our explicit formula shows that this is not the case.

As we pointed out in the Introduction, the exponent appearing in (3.76) is very similar to the one obtained by eigenvalue tunneling in matrix models. This suggests that the flat coordinate $t$ is "quantized" in units of $\alpha g_s$, i.e.

$$t = N\alpha g_s, \tag{3.78}$$

so that the free energy of the one-instanton corresponds to a background in which $N$ has decreased by one unit. As we will see in the next section, an $\ell$ multi-instanton configuration will decrease $N$ by $\ell$ units (in matrix models, $N$ is the number of eigenvalues.) The "quantization" of the

flat coordinate (3.78) is an ingredient of large $N$ dualities of the topological string [22, 30, 31], where the flat coordinate $t$ is interpreted as a 't Hooft parameter, but here it appears as a consequence solely of the HAE. The form of (3.76) is also reminiscent of the "grand partition function" for topological strings considered e.g. in [29], in which one has to sum the topological string partition function over all possible shifts of the Kähler parameters. Note however that the prefactor appearing in (3.76) is more difficult to interpret in the context of eigenvalue tunneling, and this remains an interesting problem for the future.

**Remark 3.1.** It is instructive to compare the structure of the one-instanton amplitude obtained here with the one-instanton amplitude for the NS free energy considered in [23, 32]. In the NS case one simply has

$$\exp\left(-\mathcal{G}_{\mathrm{NS}}/\hbar\right) \tag{3.79}$$

where $\mathcal{G}_{\mathrm{NS}}$ is the NS counterpart of (3.27). Its holomorphic limit is simply

$$\mathcal{G}_{\mathrm{NS}} \to \alpha\hbar\partial_t F^{\mathrm{NS}}(t,\hbar), \tag{3.80}$$

where

$$F^{\mathrm{NS}}(t,\hbar) = \sum_{n\geq 0} F_n^{\mathrm{NS}}(t)\hbar^{2n-1} \tag{3.81}$$

is the perturbative series for the NS free energy. In the NS case, the exponent appearing in the instanton amplitude does not involve a difference operator acting on the free energy, as in (3.76), but a differential operator. Note also that the prefactor of (3.76) is absent in the NS case.

### 3.4 Generalization to multi-instantons

Let us now consider multi-instantons. In principle, we have to solve the equation (3.21). However, to do this it is better to consider the partition function, instead of the free energies. The reason it that, as noted in [16], the HAE for the partition function is linear, and much easier to solve.

To proceed, let us define the perturbative partition function as

$$Z^{(0)} = \mathrm{e}^{F^{(0)}}, \tag{3.82}$$

while the full non-perturbative partition function will be denoted by $Z = \mathrm{e}^F$. We will also introduce the "reduced" partition function

$$Z_r = \frac{Z}{Z^{(0)}}. \tag{3.83}$$

We now use the fact that both $\log Z$ and $\log Z^{(0)}$ satisfy the master equation (2.22). After some easy algebra we find the equation

$$\frac{\partial Z_r}{\partial S} = \frac{g_s^2}{2}\mathcal{D}_z\mathfrak{D}_z Z_r + \mathfrak{D}_z\widetilde{F}^{(0)}\mathfrak{D}_z Z_r, \tag{3.84}$$

which is linear, as advertised.

We now solve this equation with a trans-series ansatz. We could use an ansatz mimicking (3.4), but as noted in [12, 13] we should consider a more general ansatz. The reason is the following. The topological string free energy is a formal power series in $g_s^2$, and therefore the singularities of its Borel transform (which correspond to instanton actions) come in pairs. This means that if we have a trans-series solution with an exponential of the form $\mathrm{e}^{-\mathcal{A}/g_s}$, there should

be an "anti-instanton" amplitude involving the opposite exponential $\mathrm{e}^{\mathcal{A}/g_s}$. Perhaps the simplest incarnation of this phenomenon occurs in the Painlevé I equation describing 2d gravity. The general trans-series solution to this equation was studied in [43] and it involves both instantons and "anti-instantons," as well as mixed sectors (see [44–46] for further studies of this type of trans-series). We will then assume the following ansatz for the reduced partition function,

$$Z_r = 1 + \sum_{n,m\geq 0,(n,m)\neq(0,0)} \mathcal{C}_1^n \mathcal{C}_2^m Z_r^{(n|m)}, \tag{3.85}$$

where the behavior at small $g_s$ given by

$$Z_r^{(n|m)} \sim \exp\left(-\frac{n-m}{g_s}\mathcal{A}\right). \tag{3.86}$$

The conventional multi-instanton sectors are recovered when $m = 0$:

$$Z_r^{(n)} \equiv Z_r^{(n|0)}, \qquad n \geq 0. \tag{3.87}$$

Let us note that the free energies can be easily obtained from the reduced partition functions by simply taking the logarithm. For example, we have

$$F^{(2)} = Z_r^{(2)} - \frac{1}{2}\left(Z_r^{(1)}\right)^2,$$
$$F^{(1|1)} = Z_r^{(1|1)} - Z_r^{(1|0)}Z_r^{(0|1)}. \tag{3.88}$$

We can now specialize (3.84) to the $(n|m)$ sector and use the operators $\mathsf{W}, \mathsf{D}$ to obtain

$$\mathsf{W}Z_r^{(n|m)} = \frac{g_s^2}{2}\mathsf{D}^2 Z_r^{(n|m)}. \tag{3.89}$$

Our goal is to find solutions to (3.89) with the exponential behavior (3.86). In addition, we will need boundary conditions, as we discussed in the case of the one-instanton amplitude. Boundary conditions are obtained by evaluating $Z_r^{(n|m)}$ in the frame defined by $\mathcal{A}$, like before. For reasons which will become clear later, we consider a general boundary condition of the form

$$\mathcal{Z}_{r,\mathcal{A}}^{(n|m)} = \left(\sum_{k\geq 0} a_k \mathcal{A}^k\right) \exp\left(-\frac{n-m}{g_s}\mathcal{A}\right), \tag{3.90}$$

where the coefficients $a_k$ depend only on $g_s$.

We will now write down an ansatz to solve the equation (3.89). Let us first define

$$\Phi_n = \mathsf{O}_n\mathcal{G} \tag{3.91}$$

where $\mathsf{O}_n$ is the operator

$$\mathsf{O}_n = \frac{1}{g_s\mathsf{D}}\left(1 - \mathrm{e}^{-ng_s\mathsf{D}}\right). \tag{3.92}$$

The ansatz is

$$Z_r^{(n|m)} = \mathfrak{a}_{(n|m)}\mathrm{e}^{-\Phi_{(n|m)}/g_s}, \tag{3.93}$$

where

$$\Phi_{(n|m)} = \Phi_{n-m}. \tag{3.94}$$

This already guarantees the behavior (3.86). The function $\Phi_{(n|m)}$ satisfies the same equation than $\Phi$ in (3.44), namely

$$\mathsf{W}\Phi_{(n|m)} = \frac{g_s^2}{2}\mathsf{D}^2\Phi_{(n|m)} - \frac{g_s}{2}\left(\mathsf{D}\Phi_{(n|m)}\right)^2, \tag{3.95}$$

and the proof is identical. By using (3.40) and (3.95), we find that the prefactor in (3.93) satisfies the linear equation

$$\mathsf{M}\mathfrak{a}_{(n|m)} = 0 \tag{3.96}$$

where we have defined the operator

$$\mathsf{M} = \mathsf{W} + g_s\mathsf{D}\Phi_{(n|m)}\mathsf{D} - \frac{g_s^2}{2}\mathsf{D}^2. \tag{3.97}$$

A subindex $(n|m)$ is implicit in $\mathsf{M}$, but when no confusion arises we will not write it down. In addition, $\mathfrak{a}_{(n|m)}$ satisfies the boundary condition

$$\mathfrak{a}_{(n|m),\mathcal{A}} = \sum_{k\geq 0} a_k\mathcal{A}^k. \tag{3.98}$$

Since the equation (3.96) is linear, it suffices to solve the case

$$\mathfrak{a}_{(n|m),\mathcal{A}} = \mathcal{A}^\ell, \tag{3.99}$$

for arbitrary $\ell \geq 1$. Let us now introduce the formal power series

$$X = \mathcal{G} - g_s\mathsf{D}\Phi_{(n|m)}. \tag{3.100}$$

There is again an $(n|m)$ subindex implicit in $X$. By using the same argument that we used in the one-instanton case for (3.70), it is easy to see that

$$\mathsf{M}X = 0. \tag{3.101}$$

We claim now that the solution to the problem (3.96), (3.99) is given as follows. Let us consider the set partitions of $\ell$. These can be labelled by vectors $\boldsymbol{k} = (k_1, k_2, \cdots)$ satisfying

$$d(\boldsymbol{k}) = \ell, \tag{3.102}$$

where

$$d(\boldsymbol{k}) = \sum_j jk_j. \tag{3.103}$$

Let us associate to each vector $\boldsymbol{k}$ the object

$$\mathfrak{X}_{\boldsymbol{k}} = X^{k_1}(\mathsf{D}X)^{k_2}(\mathsf{D}^2X)^{k_3}\cdots. \tag{3.104}$$

Then, the solution to the problem (3.96), (3.99) is

$$\mathfrak{w}_\ell = \sum_{\boldsymbol{k},\, d(\boldsymbol{k})=\ell} g_s^{2(\ell-|\boldsymbol{k}|)}C_{\boldsymbol{k}}\mathfrak{X}_{\boldsymbol{k}}, \tag{3.105}$$

where

$$|\boldsymbol{k}| = \sum_j k_j \tag{3.106}$$

and

$$C_{\boldsymbol{k}} = \frac{\ell!}{\prod_{j \geq 1} k_j! (j!)^{k_j}}. \tag{3.107}$$

Note that, since $\mathsf{D}$ acts as zero in the frame defined by $\mathcal{A}$, one has

$$\mathfrak{w}_{\ell,\mathcal{A}} = \mathcal{A}^\ell, \tag{3.108}$$

which is the correct boundary condition. It is less obvious that

$$\mathsf{M}\mathfrak{w}_\ell = 0, \qquad \ell \in \mathbb{Z}_{>0}. \tag{3.109}$$

Before proving this statement, let us write down some examples of (3.105):

$$\begin{aligned}
\mathfrak{w}_2 &= X^2 + g_s^2 \mathsf{D} X, \\
\mathfrak{w}_3 &= X^3 + 3g_s^2 X \mathsf{D} X + g_s^4 \mathsf{D}^2 X.
\end{aligned} \tag{3.110}$$

The key to prove (3.109) lies in the properties of the operator $\mathsf{M}$. Note that, due to the presence of $\mathsf{D}^2$, this operator is not a derivation: acting on products, we have

$$\mathsf{M}(fg) = \mathsf{M}(f)g + f\mathsf{M}(g) - g_s^2 \mathsf{D} f \mathsf{D} g. \tag{3.111}$$

However, it satisfies the useful commutation relation

$$[\mathsf{M}, \mathsf{D}] = \mathsf{D} X \mathsf{D}. \tag{3.112}$$

By using this relation and (3.101) it is easy to verify by hand that the very first $\mathfrak{w}_\ell$ in (3.110) satisfy (3.109). Proving (3.109) for all $\ell \in \mathbb{Z}_{>0}$ is equivalent to proving that

$$\mathsf{M}\Xi(\xi) = 0, \qquad \Xi(\xi) = \sum_{\ell \geq 0} \frac{\mathfrak{w}_\ell}{\ell!} \xi^\ell, \tag{3.113}$$

where we set $\mathfrak{w}_0 = 1$. A simple exercise in combinatorics shows[1]

$$\Xi(\xi) = \mathrm{e}^{\mathcal{L}_\xi X} \tag{3.114}$$

where

$$\mathcal{L}_\xi = \sum_{j=1}^{\infty} \frac{\xi^j}{j!} g_s^{2(j-1)} \mathsf{D}^{j-1} = \frac{1}{g_s^2 \mathsf{D}} \left( \mathrm{e}^{\xi g_s^2 \mathsf{D}} - 1 \right) = \frac{1}{g_s^2} \int_0^{g_s^2 \xi} \mathrm{e}^{u\mathsf{D}} \mathrm{d}u. \tag{3.115}$$

$\mathcal{L}_\xi$ is of course very similar to the operator $\mathsf{O}$ defined in (3.53). It is easy to see that (3.113) holds if and only if

$$\mathsf{M}\mathcal{L}_\xi X = \frac{g_s^2}{2} \left( \mathsf{D}\mathcal{L}_\xi X \right)^2. \tag{3.116}$$

This can be proved similarly to what we did to establish (3.44). We first have

$$\mathsf{M}\mathcal{L}_\xi = \mathcal{L}_\xi \mathsf{M} - \frac{1}{g_s^2} \int_0^{\xi g_s^2} \mathrm{d}u \, \mathrm{e}^{u\mathsf{D}} \left[ (\mathrm{e}^{-u\mathsf{D}} - 1) X \right] \mathsf{D}. \tag{3.117}$$

---

[1]This is the same exercise which is performed when one goes from the canonical to the grand canonical formalism in the cluster expansion of classical statistical mechanics, see e.g. [47], section 10.1.

Acting on $X$ and using that $\mathsf{M}(X) = 0$, we find

$$\mathsf{M}\mathcal{L}_\xi X = \frac{1}{g_s^2} \int_0^{\xi g_s^2} \mathrm{d}u \left[\left(\mathrm{e}^{u\mathsf{D}} - 1\right) X\right] \mathrm{e}^{u\mathsf{D}}\mathsf{D}X \tag{3.118}$$

On the other hand, we have that

$$(\mathsf{D}\mathcal{L}_\xi X)^2 = \frac{2}{g_s^4} \int_0^{\xi g_s^2} \left[\mathrm{e}^{u\mathsf{D}}\mathsf{D}X\right] \left[(\mathrm{e}^{u\mathsf{D}} - 1)X\right] \mathrm{d}u, \tag{3.119}$$

and (3.116) follows.

Let us make a list of observations on the above result.

1. The holomorphic limit of $\Phi_{(n|m)}$ is simply the tunneling exponent

$$g_s \left(\mathcal{F}(t) - \mathcal{F}(t - (n - m)\alpha g_s)\right), \tag{3.120}$$

and in particular, for $m = 0$ we find a tunneling of $n$ eigenvalues, as anticipated above. The general case in which $m \neq 0$ can be interpreted as tunneling of $n$ eigenvalues on the physical sheet of the mirror curve, and of $m$ eigenvalues on the non-physical sheet, as it has been proposed recently in [48]. We also note that the holomorphic limit of $X_{(m|n)}$ is

$$\alpha g_s^2 \partial_t \mathcal{F}(t - (n - m)\alpha g_s). \tag{3.121}$$

2. The solution for $Z_r^{(m|n)}$ has some symmetry properties as we change the sign of $g_s$. It follows from its definition that

$$\Phi_{(n|m)}(-g_s) = -\Phi_{(m|n)}(g_s). \tag{3.122}$$

By looking at the equation satisfied by $\mathfrak{a}_{(n|m)}$, we deduce that

$$\mathfrak{a}_{(n|m)}(-g_s) = \mathfrak{a}_{(m|n)}(g_s), \tag{3.123}$$

and we conclude that

$$Z_r^{(n|m)}(-g_s) = Z_r^{(m|n)}(g_s). \tag{3.124}$$

3. In solving for the prefactor $\mathfrak{a}_{(n|m)}$ we have constructed a correspondence

$$\mathcal{A}^\ell \to \mathfrak{w}_\ell = \mathcal{A}^\ell + \mathcal{O}(g_s), \tag{3.125}$$

which can be regarded as a quantum deformation.

So far we have considered the generic boundary condition (3.90). The results of [12, 13], as well as of this paper, indicate that there is a multi-parameter family of boundary conditions of the form (3.90) which is relevant to the resurgent structure of the topological string. This family is defined by a set of coefficients $\tau_k$, $k = 1, 2, \cdots$, and is given by

$$\mathcal{F}_{\mathcal{A}}^{(k|0)} = \tau_k \left(\frac{\mathcal{A}}{k} + \frac{g_s}{k^2}\right) \mathrm{e}^{-k\mathcal{A}/g_s}, \qquad \mathcal{F}_{\mathcal{A}}^{(0|k)} = \tau_k \left(\frac{\mathcal{A}}{k} - \frac{g_s}{k^2}\right) \mathrm{e}^{k\mathcal{A}/g_s}. \tag{3.126}$$

The mixed sectors vanish in the frame defined by $\mathcal{A}$. This family of boundary conditions is suggested by the behavior (3.18). The corresponding boundary conditions for the reduced partition functions can be obtained by exponentiation, and they are indeed of the form (3.90). One can consider a further specialization of the above family, labelled by a discrete positive integer $\ell \in \mathbb{Z}_{>0}$, in which

$$\tau_k = \delta_{k\ell}. \tag{3.127}$$

We will denote this family by the subindex $\ell$, as $F_\ell^{(n|m)}$ or $Z_{r,\ell}^{(n|m)}$ (they of course depend on the choice of instanton action $\mathcal{A}$, but we will not indicate this dependence explicitly). In this case, the boundary conditions for the reduced partition functions can be written very explicitly. When the instanton and anti-instanton numbers are both multiples of $\ell$, one has

$$\mathcal{Z}_{r,\ell,\mathcal{A}}^{(n\ell|m\ell)} = \frac{1}{n!m!} \left( \frac{\mathcal{A}}{\ell} + \frac{g_s}{\ell^2} \right)^n \left( \frac{\mathcal{A}}{\ell} - \frac{g_s}{\ell^2} \right)^m \mathrm{e}^{-\ell(n-m)\mathcal{A}/g_s}, \tag{3.128}$$

otherwise they vanish. The relevance of these boundary conditions will be explained in more detail in the next sections.

In this section we have found explicit expressions for a wide class of multi-instanton amplitudes, associated to generic boundary conditions of the form (3.90). These include all solutions considered in [12, 13]. As we will explain later, it is likely that in actual examples only the families $F_\ell^{(n|m)}$ are relevant. Let us now give some concrete examples of solutions to illustrate our general construction.

We first consider solutions in the family $Z_{r,1}^{(n|m)}$. When $n = 2, m = 0$, our general construction gives

$$Z_{r,1}^{(2)} = \frac{1}{2} \left( (X_2 + g_s)^2 + g_s^2 \mathsf{D} X_2 \right) \mathrm{e}^{-\Phi_2/g_s}, \tag{3.129}$$

where $X_2$ is defined in (3.100) and we have indicated the subindex explicitly. The holomorphic limit of this object is

$$\mathcal{Z}_{r,1}^{(2)} = \frac{1}{2} \left[ \left( g_s + \alpha g_s^2 \left( \partial_t \mathcal{F} \right) (t - 2\alpha g_s) \right)^2 + \alpha^2 g_s^4 \partial_t^2 \mathcal{F}(t - 2\alpha g_s) \right] \mathrm{e}^{\mathcal{F}(t - 2\alpha g_s) - \mathcal{F}(t)}. \tag{3.130}$$

Another interesting case is $m = n = 1$, where one finds

$$Z_{r,1}^{(1|1)} = \mathcal{G}^2 - g_s^2 + g_s^2 \mathsf{D}\mathcal{G}. \tag{3.131}$$

Its holomorphic limit is

$$\mathcal{Z}_{r,1}^{(1|1)} = \alpha^2 g_s^4 \left[ (\partial_t \mathcal{F})^2 + \partial_t^2 \mathcal{F} \right] - g_s^2. \tag{3.132}$$

Finally, let us consider the first non-trivial example in the family with $\ell \geq 2$:

$$Z_{r,\ell}^{(\ell)} = \left( \frac{X_\ell}{\ell} + \frac{g_s}{\ell^2} \right) \mathrm{e}^{-\Phi_\ell/g_s}. \tag{3.133}$$

Its holomorphic limit is simply

$$\mathcal{Z}_{r,\ell}^{(\ell)} = \left( \frac{\alpha g_s^2}{\ell} \partial_t \mathcal{F}(t - \ell \alpha g_s) + \frac{g_s}{\ell^2} \right) \mathrm{e}^{\mathcal{F}(t - \ell \alpha g_s) - \mathcal{F}(t)}. \tag{3.134}$$

**Remark 3.2.** Note that the reduced partition function $Z_{r,\ell}^{(\ell)}$ is equal to the free energy $F_\ell^{(\ell)}$, since $F_\ell^{(\ell')} = 0$ for $\ell' < \ell$. The two-instanton amplitude $Z_{r,2}^{(2)}$ already appeared in [13], but it led to some confusions there since it does not solve the same equation than $F_1^{(2)}$. In our formulation, this is due to the fact that they satisfy different boundary conditions.

## 3.5 From multi-instantons to the resurgent structure

As in the theory of nonlinear ODEs, the trans-series obtained from the HAE are expected to be the building blocks for the resurgent structure of the topological string. The resurgent structure of a given perturbative series can be formalized in many different ways, but in this paper the most convenient way consists of specifying the so-called *alien derivatives* of the different formal series appearing in the theory (alien calculus was originally developed by Écalle in [49], and accessible introductions can be found in [3–5]).

One remarkable property of the trans-series solutions that we have obtained is that they are all constructed from the perturbative series and its derivatives. This means that, if we know the alien derivatives of the perturbative series $\mathcal{F}^{(0)}$, we can deduce in principle the alien derivatives of all the multi-instanton trans-series. In this section we will obtain expressions for all these alien derivatives, for the family of multi-instanton solutions introduced above and denoted by $\mathcal{F}_\ell^{(n|m)}$. Let us briefly review some ingredients of the theory of resurgence which we will need in the following.

Given a formal Gevrey-1 power series

$$\varphi(z) = \sum_{n \geq 0} a_n z^n, \tag{3.135}$$

its Borel transform is defined by

$$\widehat{\varphi}(\zeta) = \sum_{n \geq 0} \frac{a_n}{n!} \zeta^n. \tag{3.136}$$

We will assume that $\varphi(z)$ are resurgent functions. This means essentially that their Borel transforms can be analytically continued across the complex plane except for a discrete set of singularities.

Let us now fix a value $z$, and let $\theta = \arg z$. If $\widehat{\varphi}(\zeta)$ analytically continues to an integrable function along the ray $\mathcal{C}^\theta := \mathrm{e}^{\mathrm{i}\theta} \mathbb{R}_+$, the Borel resummation of $\varphi(z)$ is given by the Laplace transform

$$s(\varphi)(z) = \int_0^\infty \widehat{\varphi}(z\zeta) \mathrm{e}^{-\zeta} \mathrm{d}\zeta = \frac{1}{z} \int_{\mathcal{C}^\theta} \widehat{\varphi}(\zeta) \mathrm{e}^{-\zeta/z} \mathrm{d}\zeta. \tag{3.137}$$

Let $\zeta_\omega$ be a singularity of $\widehat{\varphi}(\zeta)$. A ray in the $\zeta$-plane which starts at the origin and passes through $\zeta_\omega$ is called a *Stokes ray*. It is of the form $\mathrm{e}^{\mathrm{i}\theta} \mathbb{R}_+$, where $\theta = \arg(\zeta_\omega)$. Clearly, Borel resummations are not defined along Stokes rays, but one can define instead *lateral* resummations as follows. Let $\mathcal{C}_\pm^\theta$ be contours starting at the origin and going slightly above (respectively, below) the Stokes ray $\mathcal{C}^\theta$. Then, the lateral resummations are defined by

$$s_\pm(\varphi)(z) = \frac{1}{z} \int_{\mathcal{C}_\pm^\theta} \widehat{\varphi}(\zeta) \mathrm{e}^{-\zeta/z} \mathrm{d}\zeta. \tag{3.138}$$

Due to the presence of singularities, the two lateral resummations differ in exponentially small corrections. Let us denote by $\Omega_\theta$ the set indexing the singularities along the ray $\mathcal{C}^\theta$. Then, one has the discontinuity formula

$$s_+(\varphi)(z) - s_-(\varphi)(z) = \mathrm{i} \sum_{\omega \in \Omega_\theta} \mathsf{s}_\omega \, \mathrm{e}^{-\zeta_\omega/z} s_-(\varphi_\omega)(z), \tag{3.139}$$

where $\varphi_\omega(z)$ is a formal power series associated to the singularity $\zeta_\omega$ of $\widehat{\varphi}(\zeta)$. Given a choice of normalization for these series, the discontinuity relation (3.139) defines non-trivial *Stokes constants* $\mathsf{s}_\omega$.

The result (3.139) involves Borel resummed formal series, but it is useful to rewrite it as a relation between formal series themselves. If we regard lateral Borel resummations as operators, we introduce the *Stokes automorphism* along the ray $\mathcal{C}^\theta$, $\mathfrak{S}_\theta$, as

$$s_{+\theta} = s_{-\theta}\mathfrak{S}_\theta. \tag{3.140}$$

Then, we can write (3.139) as

$$\mathfrak{S}_\theta(\varphi) = \varphi + \mathrm{i} \sum_{\omega \in \Omega_\theta} \mathsf{s}_\omega \mathrm{e}^{-\zeta_\omega/z} \varphi_\omega. \tag{3.141}$$

We define now the (pointed) alien derivative $\dot{\Delta}_{\zeta_\omega}$ associated to the singularity $\zeta_\omega$, $\omega \in \Omega_\theta$, by

$$\mathfrak{S}_\theta = \exp\left(\sum_{\omega \in \Omega_\theta} \dot{\Delta}_{\zeta_\omega}\right). \tag{3.142}$$

The most important property of alien derivatives is that they are indeed derivatives, i.e. they satisfy Leibniz rule when acting on a product of formal series:

$$\dot{\Delta}_{\zeta_\omega}\left(\phi_1(z)\phi_2(z)\right) = \left(\dot{\Delta}_{\zeta_\omega}\phi_1(z)\right)\phi_2(z) + \phi_1(z)\left(\dot{\Delta}_{\zeta_\omega}\phi_2(z)\right). \tag{3.143}$$

Given now a formal power series $\varphi(z)$ as a starting point, we can iterate the above procedure to eventually find a "complete" set of formal power series

$$\mathfrak{B}_\varphi = \{\varphi_\omega(z)\}_{\omega \in \Omega}, \tag{3.144}$$

labelled by a set $\Omega$, in such a way that the operation of the alien derivatives closes in this set:

$$\dot{\Delta}_{\mathcal{A}}\varphi_\omega = \sum_{\omega'} \mathsf{S}^{\mathcal{A}}_{\omega\omega'}\varphi_{\omega'}. \tag{3.145}$$

The coefficients $\mathsf{S}^{\mathcal{A}}_{\omega\omega'}$ can be obtained from the Stokes coefficients appearing in the discontinuities, see e.g. [4] for additional examples and clarifications. We call the set $\mathfrak{B}_\varphi$, together with the action of all alien derivatives on it, the (minimal) *resurgent structure* associated to $\varphi$ [50]. This notion is a mathematical formulation of the non-perturbative information that can be obtained from a perturbative series $\varphi(z)$.

We would like to understand the resurgent structure associated to the topological string perturbation series. This is a well-defined problem (under the mild assumption that the series involved are resurgent), and it provides a concrete characterization of the non-perturbative structure of the theory. In addition, it was conjectured in [50, 51] that the Stokes coefficients appearing in this resurgent structure are interesting invariants counting BPS states. In the case of the resolved conifold, the resurgent structure was studied in [41], and more recently it was studied from the point of view of the relation to BPS counting in [52–54]. Of course, the resolved conifold is in many ways too simple an example, and life starts being interesting for non-trivial toric CYs with one modulus, like those studied in [13] and in the present paper. We will now make various proposals for the resurgent structure of the topological string in this case, based on the results above and on some numerical calculations described in the next section.

Let $\mathcal{F}_g^{(0)}(t)$ be the holomorphic free energies at genus $g$, in a given frame. The Borel transform

$$\widehat{\mathcal{F}}^{(0)}(t, \zeta) = \sum_{g \geq 0} \frac{1}{(2g)!} \mathcal{F}_g^{(0)}(t) \zeta^{2g} \tag{3.146}$$

will have singularities filling a subset of a lattice in the complex plane. Let us consider a singularity $\ell \mathcal{A}$, where $\mathcal{A}$ is a "primitive" singularity of the form (3.2), and $\ell \in \mathbb{Z}_{>0}$ is a positive integer. There are two different cases to consider. If $\alpha = 0$, i.e. if the instanton action is given by the flat coordinate of the frame (up to a linear shift by a constant), then the multi-instanton trans-series are trivial, and of the form (3.18). In this case, we will have

$$\dot{\Delta}_{\ell \mathcal{A}} \mathcal{F}^{(0)} = \mathsf{S}_{\ell \mathcal{A}}(g_s) \mathcal{F}_{\mathcal{A}}^{(\ell)}. \tag{3.147}$$

Here, $\mathsf{S}_{\ell \mathcal{A}}(g_s)$ is a Stokes coefficient which depends on $g_s$ and in principle also on the modulus $t$. Our concrete calculations indicate that the dependence on $g_s$ is simple and that they are locally constant functions of $t$. We expect the Stokes coefficients to be independent of $\ell$ in many cases. This is suggested by the large order behavior (3.15).

Let us now consider the more interesting case $\alpha \neq 0$, in which the multi-instantons take the more complicated form discussed in the previous sections. We conjecture the following result for the pointed alien derivatives,

$$\dot{\Delta}_{\ell \mathcal{A}} \mathcal{F}^{(0)} = \mathsf{S}_{\ell \mathcal{A}}(g_s) \mathcal{F}_{\ell}^{(\ell)}, \tag{3.148}$$

where $\mathcal{F}_{\ell}^{(\ell)}$ is the holomorphic limit of the multi-instanton amplitude $F_{\ell}^{(n|m)}$ with $n = \ell$, $m = 0$, corresponding to the family of solutions (3.127). $\mathsf{S}_{\ell \mathcal{A}}(g_s)$ is a Stokes coefficient with the properties noted above, and in particular we expect it to be independent of $\ell$ in many situations, as in (3.147). Since $\mathcal{F}^{(0)}$ is an even power series in $g_s$, we have

$$\dot{\Delta}_{-\ell \mathcal{A}} \mathcal{F}^{(0)} = \mathsf{S}_{\ell \mathcal{A}}(-g_s) \mathcal{F}_{\ell}^{(0|\ell)}. \tag{3.149}$$

The conjecture (3.148) contains and extends empirical results obtained in [13] and further developed in the next section. We note that a similar conjecture applies to the NS free energy [32]. The Stokes coefficients encode information about the resurgent structure of the theory and they are non-trivial. Currently, we can only calculate them numerically, and we have access to very few of them.

We should think about (3.148) as the "primitive" alien derivatives, from which additional alien derivatives can be calculated. This is simply because all multi-instantons are functionals of $\mathcal{F}^{(0)}$ itself, and the action of alien derivatives commutes with taking derivatives w.r.t. $t$. We then have the following formula for the full family of trans-series sectors $\mathcal{F}_{\ell}^{(n|m)}$:

$$\dot{\Delta}_{\ell \mathcal{A}} \mathcal{F}_{\ell}^{(\ell n | \ell m)} = \mathsf{S}_{\ell \mathcal{A}}(g_s)(n+1) \mathcal{F}_{\ell}^{(\ell(n+1)|\ell m)}, \tag{3.150}$$

from which one obtains

$$\dot{\Delta}_{-\ell \mathcal{A}} \mathcal{F}_{\ell}^{(\ell n | \ell m)} = \mathsf{S}_{\ell \mathcal{A}}(-g_s)(m+1) \mathcal{F}_{\ell}^{(\ell n | \ell(m+1))}. \tag{3.151}$$

Let us emphasize that (3.150) follows directly from (3.148) by taking derivatives, and although we don't have a proof of the general formula, we have checked it in many cases.

**Example 3.3.** Let us consider the case $\ell = 1$. The one-instanton amplitude is given in (3.76),

$$\mathcal{F}_1^{(1)} = \left( g_s + \alpha g_s^2 \left( \partial_t \mathcal{F} \right) \left( t - \alpha g_s \right) \right) \mathrm{e}^{\mathcal{F}(t - \alpha g_s) - \mathcal{F}(t)}. \tag{3.152}$$

One finds, by a direct calculation,

$$\begin{aligned} \dot{\Delta}_{\mathcal{A}} \mathcal{F}_1^{(1)} &= \alpha g_s^2 \partial_t \left( \dot{\Delta}_{\mathcal{A}} \mathcal{F} \right) \left( t - \alpha g_s \right) \mathrm{e}^{\mathcal{F}(t - \alpha g_s) - \mathcal{F}(t)} \\ &\quad + \left( \dot{\Delta}_{\mathcal{A}} F(t - \alpha g_s) - \dot{\Delta}_{\mathcal{A}} F(t) \right) \left( g_s + \alpha g_s^2 \left( \partial_t \mathcal{F} \right) \left( t - \alpha g_s \right) \right) \mathrm{e}^{\mathcal{F}(t - \alpha g_s) - \mathcal{F}(t)}, \end{aligned} \tag{3.153}$$

and by using (3.148) one obtains

$$\begin{aligned} \dot{\Delta}_{\mathcal{A}} \mathcal{F}_1^{(1)} &= \mathsf{S}_{\mathcal{A}}(g_s) \Bigg\{ \left[ \left( g_s + \alpha g_s^2 \left( \partial_t \mathcal{F} \right) \left( t - 2\alpha g_s \right) \right)^2 + \alpha^2 g_s^4 \partial_t^2 \mathcal{F}(t - 2\alpha g_s) \right] \mathrm{e}^{\mathcal{F}(t - 2\alpha g_s) - \mathcal{F}(t)} \\ &\quad - \left( \mathcal{F}_1^{(1)} \right)^2 \Bigg\}. \end{aligned} \tag{3.154}$$

If we take into account the explicit expression (3.130), we conclude that

$$\dot{\Delta}_{\mathcal{A}} \mathcal{F}_1^{(1)} = S_{\mathcal{A}}(g_s) \, 2 \, \mathcal{F}_1^{(2)}, \tag{3.155}$$

in agreement with (3.150) for $\ell = n = 1$, $m = 0$. $\qquad\square$

The result for the alien derivatives in (3.150) is very similar to what one obtains in nonlinear ODEs with the help of the so-called bridge equation (see e.g. [3–5]). This can be understood as follows. Let us define the multi-instanton free energy

$$F_\ell^{\mathrm{np}} = \sum_{n,m \geq 0, (n,m) \neq (0,0)} \mathcal{C}_1^n \mathcal{C}_2^m F_\ell^{(\ell n | \ell m)} \tag{3.156}$$

which contains all the non-perturbative trans-series associated to the solution characterised by (3.127). This free energy satisfies the non-linear equation obtained from the HAE

$$\mathsf{W} F_\ell^{\mathrm{np}} = \frac{g_s^2}{2} \left( \mathsf{D}^2 F_\ell^{\mathrm{np}} + \left( \mathsf{D} F_\ell^{\mathrm{np}} \right)^2 \right). \tag{3.157}$$

Since the dotted alien derivative commutes with the operators $\mathsf{W}$, $\mathsf{D}$, one obtains the linearized equation

$$\mathsf{W} \left( \dot{\Delta}_{\ell \mathcal{A}} F_\ell^{\mathrm{np}} \right) = \frac{g_s^2}{2} \left( \mathsf{D}^2 \left( \dot{\Delta}_{\ell \mathcal{A}} F_\ell^{\mathrm{np}} \right) + 2 \mathsf{D} F_\ell^{\mathrm{np}} \mathsf{D} \left( \dot{\Delta}_{\ell \mathcal{A}} F_\ell^{\mathrm{np}} \right) \right). \tag{3.158}$$

The same equation is satisfied by the derivatives

$$\frac{\partial F_\ell^{\mathrm{np}}}{\partial \mathcal{C}_1}, \qquad \frac{\partial F_\ell^{\mathrm{np}}}{\partial \mathcal{C}_2} \tag{3.159}$$

and their linear combinations. Although we don't seem to have a uniqueness result to guarantee it, it is natural to assume that

$$\dot{\Delta}_{\ell \mathcal{A}} F_\ell^{\mathrm{np}} = \mathsf{S}_{\ell \mathcal{A}}(g_s) \frac{\partial F_\ell^{\mathrm{np}}}{\partial \mathcal{C}_1} + \mathsf{T}_{\ell \mathcal{A}}(g_s) \frac{\partial F_\ell^{\mathrm{np}}}{\partial \mathcal{C}_2}. \tag{3.160}$$

Indeed, the equation (3.150) says that this is the case, and that in addition $\mathsf{T}_{\ell \mathcal{A}}(g_s) = 0$.

In the next section we will illustrate the conjectures and results of this section with the example of local $\mathbb{P}^2$.

## 4 Examples and experimental evidence

The toric CY known as local $\mathbb{P}^2$ is perhaps the simplest CY manifold with a rich topological string theory, and it has been studied from many points of view since the early days of local mirror symmetry [17, 55]. In Appendix A we summarize some of the ingredients needed to analyze this model with the holomorphic anomaly equations. As it should be clear by now, this is an example of "parametric resurgence," i.e. the whole resurgent structure varies as we move on the moduli space, and we expect to have a rich structure (e.g. wall-crossing phenomena) which has not been fully unveiled yet. The moduli space of local $\mathbb{P}^2$ is parametrized by a complex variable $z$, in such a way that $z = 0$ corresponds to the large radius point, while $z = -1/27$ corresponds to the conifold point. We will mostly focus on the region in moduli space in which $z$ is real,

$$-\frac{1}{27} < z < 0 \tag{4.1}$$

which describes essentially the geometric phase of the theory. Of course, one can explore other regions.

The results in this section are based on numerical calculations of the perturbative free energies $\mathcal{F}_g(t)$ in two different frames: the large radius frame, and the conifold frame (we always work on the holomorphic limit). These calculations allow us to determine the location of the very first Borel singularities, and to estimate numerically some of the Stokes constants.

### 4.1 Borel singularities and Stokes constants in the large radius frame

Let us first consider the large radius frame. For values of $z$ close to the conifold point, we find Borel singularities at

$$\ell \mathcal{A}_c, \qquad \ell \in \mathbb{Z}, \tag{4.2}$$

where

$$\mathcal{A}_c = \frac{2\pi i}{\sqrt{3}} t_c. \tag{4.3}$$

This is the singularity (3.16) expected just from the conifold behavior (in this case, as noted in (A.18), one has $\mathfrak{b} = 3$). By using the results in Appendix A, it is easy to check that for this action, $\alpha = 3i$ in (3.2).

A good graphical guide to the location of Borel singularities is obtained as follows. One takes an approximation to the Borel transform by picking a finite number of terms in (3.146). The diagonal Padé approximant to this polynomial provides a reasonable approximation to the analytic continuation of the Borel transform, and its poles mimic the location of branch cuts. In Fig. 1 we plot these poles for $z = -1/30$. The figure on the left is made for the standard Padé approximant, showing clearly the location of the first singularity (4.2) with $\ell = 1$. The singularities with highest values of $\ell$ are hidden behind the line of poles, but one can unveil the very first ones by using a conformal Padé approximant, i.e. by combining the Padé approximant with the conformal map

$$\zeta = \frac{1}{i} \frac{2\mathcal{A}_c \xi}{1 - \xi^2}, \tag{4.4}$$

see e.g. [56] for a summary of this class of techniques in Borel analysis. The resulting plot, shown in the right, displays the singularities with $\ell = 2, 3, 4$ in the $\xi$-plane.

We can now ask what is the value of the "primary" alien derivatives associated to these singularities, or equivalently, we can calculate the discontinuity of the lateral Borel transforms

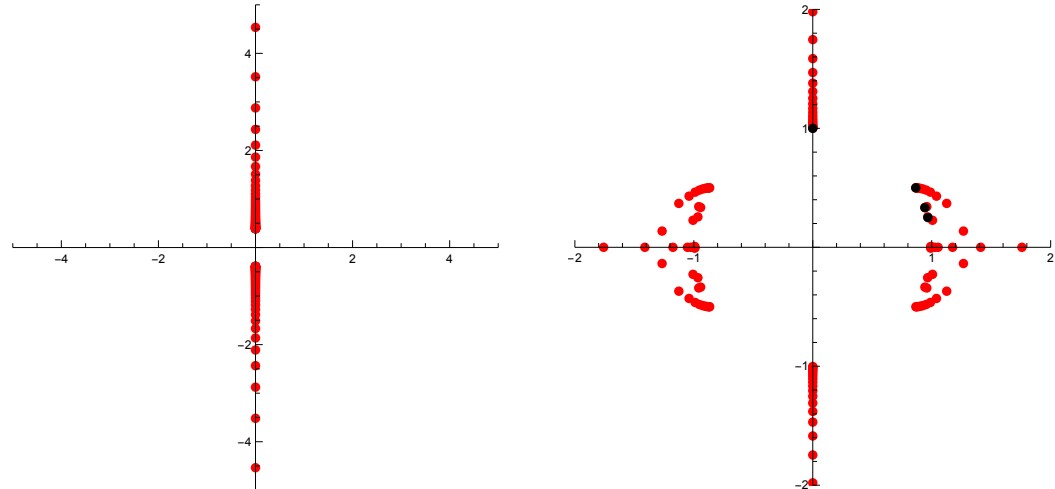

**Figure 1**. The Borel singularities for large radius free energies $\mathcal{F}_g$, at $z = -1/30$. On the left we plot the poles of the conventional Padé approximant, while on the right we plot the poles of the conformal Padé approximant in the $\xi$-plane, where $\xi$ is defined in (4.4). In both cases we use the free energies up to $g = 125$. The black dots in the plot on the right correspond to the location of the singularities (4.2) in the $\xi$-plane, with $\ell = 1, 2, 3, 4$.

across, say, the positive imaginary axis. Explicit numerical calculations up to the two instanton order show that

$$(s_+ - s_-)(\mathcal{F}^{(0)}) = s_- \left\{ \frac{\mathrm{i}}{2\pi g_s} \mathcal{F}_1^{(1)} - \frac{1}{4\pi^2 g_s^2} \mathcal{F}_1^{(2)} + \frac{\mathrm{i}}{2\pi g_s} \mathcal{F}_2^{(2)} + \cdots \right\}. \tag{4.5}$$

As a technical comment, note that the Borel resummation of the multi-instanton amplitudes involves just the resummation of the sequence of derivatives of $\mathcal{F}_g(t)$, evaluated at a shifted argument, and therefore it is straightforward. By using (3.150), we deduce that the result (4.5) is equivalent to

$$\dot{\Delta}_{\mathcal{A}_c} \mathcal{F}^{(0)} = \frac{\mathrm{i}}{2\pi g_s} \mathcal{F}_1^{(1)}, \qquad \dot{\Delta}_{2\mathcal{A}_c} \mathcal{F}^{(0)} = \frac{\mathrm{i}}{2\pi g_s} \mathcal{F}_2^{(2)}, \tag{4.6}$$

which is indeed as expected from the conjecture (3.148), and gives in addition the value

$$\mathsf{S}_{\ell \mathcal{A}_c}(g_s) = \frac{\mathrm{i}}{2\pi g_s}, \qquad \ell = 1, 2. \tag{4.7}$$

Note that the Stokes constants seem to be independent of $\ell$, as we anticipated above. These results summarize compactly many of the numerical results obtained in [13] by large order analysis. We note that, by using (3.77), one finds the explicit large order formula

$$\mathcal{F}_g \sim \frac{1}{2\pi^2} \exp\left( \frac{\alpha^2}{2} \partial_t^2 \mathcal{F}_0 \right) \mathcal{A}^{-2g+2} \Gamma(2g - 1), \qquad g \gg 1. \tag{4.8}$$

We also note that the values of the Stokes constants in (4.6) are dictated by the large order behavior at the conifold (3.15) (for example, the power $g_s^{-1}$ is due to the shift $-1$ in $\Gamma(2g - 1)$).

As we approach the large radius point at $z = 0$, we find a tower of singularities located at

$$\pm 2\pi t(z) + 4\pi^2 \mathrm{i}m, \qquad m \in \mathbb{Z}. \tag{4.9}$$

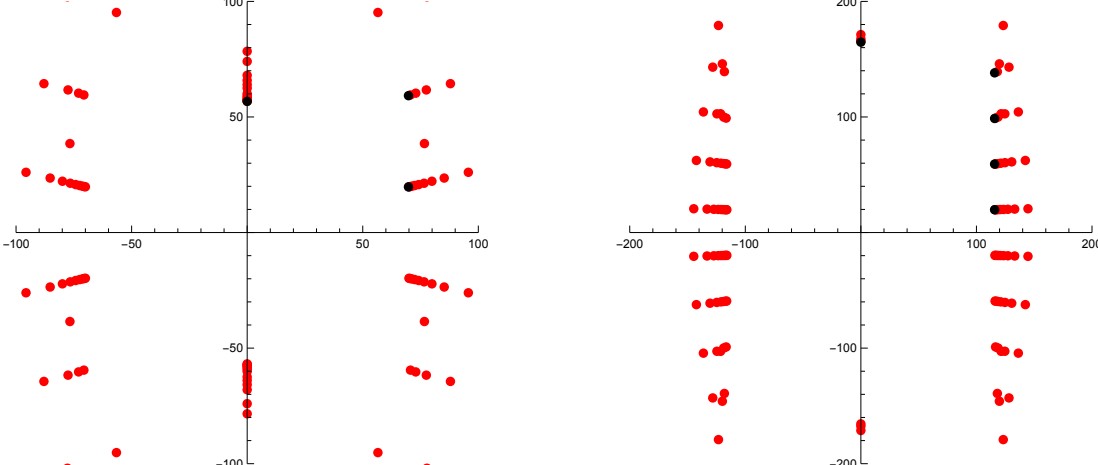

**Figure 2**. The Borel singularities for the free energies at large radius, for $z = -15 \cdot 10^{-6}$ (left) and $z = -10^{-8}$ (right). We use the poles of the Padé approximant and the free energies $\mathcal{F}_g$ with $g$ up to 120. The black dots in the first quadrant correspond to the tower $\mathcal{A}_{0,n}^+$ (4.10), for the values $n = 0, 1$ (left) and $n = 0, 1, 2, 3$ (right). The black dots in the positive imaginary axis corresponds to the location of the singularity (4.2) with $\ell = 1$.

This type of towers, also called "peacock patterns" in [57], appeared before in [19, 41], in the context of topological string theory, and also in complex Chern–Simons theory [57–59]. Note that, for negative values of $z$, the points (4.9) can be written as

$$\mathcal{A}_{0,n}^{\pm} = \pm 2\pi \mathrm{Re}(t(z)) + 4\pi^2 \mathrm{i} \left( n + \frac{1}{2} \right), \qquad n \in \mathbb{Z}. \tag{4.10}$$

(The zero subindex will become clear in (4.19)). Some of these singularities are shown in Fig. 2. Since the instanton actions (4.10) do not involve the derivative of the prepotential in this frame, the corresponding instanton amplitudes are of the "trivial" form (3.18). We expect these towers of singularities to lead however to non-trivial Stokes constants, as found in a closely related context in [50] (see also in [57–60] for similar situations in complex Chern–Simons theory). One finds, for example,

$$\dot{\Delta}_{\mathcal{A}_{0,0}^+} \mathcal{F}^{(0)} = \frac{3\mathrm{i}}{2\pi g_s} \mathcal{F}_{\mathcal{A}_{0,0}^+}^{(1)}, \tag{4.11}$$

which is equivalent to the result reported in the Appendix A of [19]. We expect to have, more generally,

$$\dot{\Delta}_{\ell \mathcal{A}_{0,n}^+} \mathcal{F}^{(0)} = \frac{\mathrm{i}}{2\pi g_s} \mathfrak{s}_{n,\ell} \mathcal{F}_{\ell \mathcal{A}_{0,n}^+}^{(\ell)}, \tag{4.12}$$

where $\mathfrak{s}_{n,\ell}$ are Stokes coefficients. It is likely that these numbers are integers (or at least rational numbers), and related to BPS invariants of some type.

We should note that the above results give just numerical approximations to the structure of singularities of the Borel plane. The most general singularity is expected to be of the form

$$2\pi k t(z) + 4\pi^2 \mathrm{i} n + \ell \mathcal{A}_c(z) \tag{4.13}$$

for integer numbers $k$, $n$, $\ell$. Our numerical results tell us that some of these points are definitely realized as actual singularities, but singularities which are sufficiently far from the origin are not detected by our analysis.

## 4.2 Borel singularities and Stokes constants in the conifold frame

Let us now consider the free energies in the conifold frame, which we will denote by $\mathcal{F}_g(t_c)$ (we do not add the superscript $c$ to make our notation less heavy, and it is understood that in this section all free energies refer to the conifold frame). In this case, one can improve the numerical analysis in various ways. First of all, one finds a sequence of singularities in the imaginary axis of the form (4.2), but in the conifold frame these are solely due to the singular part of the conifold free energy. We can then divide $\mathcal{F}_g(t_c)$ into a regular and a singular part,

$$\mathcal{F}_g(t_c) = \mathcal{F}_g^{\mathrm{sing}}(t_c) + \mathcal{F}_g^{\mathrm{reg}}(t_c), \tag{4.14}$$

where

$$\mathcal{F}_g^{\mathrm{sing}}(t_c) = 3^{g-1} \frac{B_{2g}}{2g(2g-2)} t_c^{2-2g}, \qquad g \geq 2, \tag{4.15}$$

is the polar part in (A.18), and

$$\mathcal{F}_0^{\mathrm{sing}}(\lambda) = \frac{t_c^2}{6} \log\left(\frac{t_c}{27}\right) - \frac{t_c^2}{4},$$
$$\mathcal{F}_1^{\mathrm{sing}}(\lambda) = -\frac{1}{12} \log\left(\frac{t_c}{27}\right). \tag{4.16}$$

We can then remove the singularities (4.2) by considering the regular free energies. When one does that, one uncovers additional singularities which are not numerically visible in the original free energies. One finds two towers of Borel singularities. The first one is of the form

$$\mathcal{A}_{c,n} = \mathcal{A}_c + 4\pi^2 \mathrm{i} n, \qquad n \in \mathbb{Z}. \tag{4.17}$$

The second one is of the form

$$\pm 2\pi t(z) + 4\pi^2 \mathrm{i} m + \ell \mathcal{A}_c(z), \qquad m, \ell \in \mathbb{Z} \tag{4.18}$$

and we will write it as

$$\mathcal{A}_{\ell,n}^{\pm} = \pm 2\pi \mathrm{Re}(t(z)) + 4\pi^2 \mathrm{i}\left(n + \frac{1}{2}\right) + \ell \mathcal{A}_c(z), \qquad n, \ell \in \mathbb{Z}. \tag{4.19}$$

(Note that, in contrast, in the large radius frame, the tower of singularities (4.10) has $\ell = 0$.) The leading singularities near the conifold point turn out to be at $\pm \mathcal{A}_s$, $\pm \mathcal{A}_s^\star$, where $\mathcal{A}_s = \mathcal{A}_{-1,0}^+$. We show some of these singularities in Fig. 3, by using Padé approximants.

There are two Stokes constants that can be calculated with precision. The first one corresponds to the singularity (4.17) with $n = 1$. The corresponding trans-series is of the "trivial" type and is given by (3.13):

$$\mathcal{F}_{\mathcal{A}_{c,1}}^{(1)} = \left(\mathcal{A}_c + 4\pi^2 \mathrm{i} + g_s\right) \exp\left\{-\frac{1}{g_s}(\mathcal{A}_c + 4\pi^2 \mathrm{i})\right\}. \tag{4.20}$$

We find, with high numerical precision,

$$\dot{\Delta}_{\mathcal{A}_{c,1}} \mathcal{F}^{(0)} = \frac{3\mathrm{i}}{2\pi g_s} \mathcal{F}_{\mathcal{A}_{c,1}}^{(1)}. \tag{4.21}$$

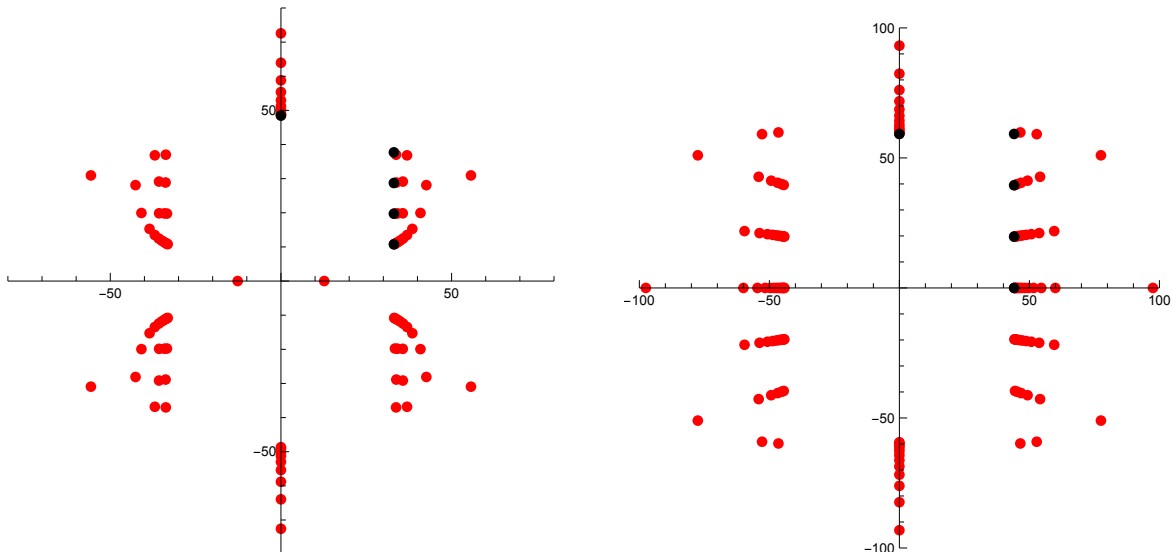

**Figure 3**. The Borel singularities for the regulated free energies in the conifold frame $\mathcal{F}_g^{\text{reg}}$, for $z = -1/200$ (left), and for the value of $z$ such that $\mathcal{A}_c(z) = 2\pi^2 \mathrm{i}$ (right). We use a Padé approximant with a maximal value of $g = 100$ in the first case and $g = 130$ in the second. The black dots in the first quadrant correspond to the singularities $\mathcal{A}_{-1,0}^+$, $\mathcal{A}_{0,0}^+$, $\mathcal{A}_{1,0}^+$, $\mathcal{A}_{2,0}^+$. The black dot in the positive imaginary axis corresponds to the location of the singularity $\mathcal{A}_{c,1}$.

We expect to have, more generally,

$$\dot{\Delta}_{\ell \mathcal{A}_{c,n}} \mathcal{F}^{(0)} = \frac{\mathrm{i}}{2\pi g_s} \mathfrak{s}_{n,\ell}^c \mathcal{F}_{\ell \mathcal{A}_{c,n}}^{(\ell)}. \tag{4.22}$$

The second Stokes constant that can be computed with precision corresponds to $\mathcal{A}_s$, at the point in moduli space where $\mathcal{A}_c = 2\pi^2 \mathrm{i}$. This is because, for that value, $\mathcal{A}_s$ is real and the discontinuity along the real direction is easier to calculate. First of all, we have to calculate the one-instanton contribution associated to $\mathcal{A}_s$. We write the action as in (3.2),

$$\mathcal{A}_s = -3^{3/2} \partial_{t_c} \mathcal{F}_0^c(t_c) - \mathrm{i} \left( \frac{2\pi\sqrt{3}}{3} t_c - 2\pi^2 \right), \tag{4.23}$$

where we used that, due to (A.17),

$$\alpha = -3^{3/2}. \tag{4.24}$$

This means that in the formulae for the non-perturbative one-instanton correction one has to use the prepotential (3.28),

$$\mathcal{F}_0^c(t_c) + \frac{a}{2} t_c^2 + b t_c, \tag{4.25}$$

where

$$a = \frac{2\pi\mathrm{i}}{9}, \qquad b = -\frac{2\pi^2\mathrm{i}}{3^{3/2}}. \tag{4.26}$$

The one-instanton correction can be written as

$$\mathcal{F}_{\mathcal{A}_s}^{(1)} = - \left\{ g_s + \alpha g_s^2 \partial_{t_c} \mathcal{F}^c(t_c - \alpha g_s) + 2\pi^2 \mathrm{i} - \mathcal{A}_c(z) - 6\pi \mathrm{i} g_s \right\}$$
$$\times \exp \left[ \mathcal{F}^c(t_c - \alpha g_s) - \mathcal{F}^c(t_c) - \frac{1}{g_s} \left( 2\pi^2 \mathrm{i} - \mathcal{A}_c(z) \right) \right]. \tag{4.27}$$

One finds, when $\mathcal{A}_c(z) = 2\pi^2 \mathrm{i}$,

$$\dot{\Delta}_{\mathcal{A}_s} \mathcal{F}^{(0)} = \frac{\mathrm{i}}{2\pi g_s} \left( \mathcal{F}^{(1)}_{\mathcal{A}_s} + \mathcal{F}^{(1)}_{\mathcal{A}_s^\star} \right), \tag{4.28}$$

since the action $\mathcal{A}_s$ and its complex conjugate come together precisely at that point in moduli space. The one-instanton amplitudes associated to $\mathcal{A}_s$, $\mathcal{A}_s^\star$ are the non-perturbative effects studied in [19] in order to compare the resurgent structure of the topological string, to the TS/ST correspondence of [20, 21].

## 5 Conclusions and open problems

In this paper we have found exact, closed form multi-instanton solutions to the trans-series extension of the BCOV equations proposed in [12, 13]. The main tool to achieve this is an operator formulation of the equations, akin to what was done in [23, 24]. Our results include all the solutions found in [13], and they generalize them to arbitrary multi-instantons. In addition, we found that the holomorphic limit of these solutions is very simple: it can be written solely in terms of the perturbative free energies and can be naturally interpreted in terms of eigenvalue tunneling. In particular, our result suggests that the flat coordinates are naturally quantized in units of the string coupling constant. This is a working hypothesis in large $N$ dualities for the topological string, but here it follows from the non-perturbative structure of the holomorphic anomaly equations.

Our results are a first step in decoding the full resurgent structure of the topological string, which requires a determination of the actual Borel singularities and their Stokes constants. We have given a first taste of these issues in section 4, but there is clearly much more to do. The calculation of Stokes constants by numerical means reaches very quickly its limits, and one should find more clever approaches to the problem. In [50] we proposed a modified version of the resurgent structure of the topological string in the conifold frame, involving only numerical power series (in contrast to the parametric power series considered in this paper). This version leads to calculable Stokes constants, and it would be very interesting to find the precise relation between the framework of [50] and the problem considered here.

As we have mentioned in section 4, we expect the Stokes constants appearing in this problem to be closely related to BPS invariants, and further evidence along this direction will be presented in [61]. A similar problem displaying this connection to BPS counting is the resurgent structure of quantum periods, studied in the companion paper [32]. However, we should note that the two problems differ in many important points. In particular, formulae like (4.5) show that, in the case of the topological string, Stokes automorphisms are more complicated than in the case of quantum periods, where they are given by the so-called Delabaere–Pham formula [62]. In that sense, the Riemann–Hilbert problem naturally associated to the topological string should be different from the one considered in e.g. [63, 64], which assumes, following [65], Stokes automorphisms of the Delabaere–Pham type. A related observation is that the resolved conifold, which is so far the only example worked out in detail in the approach of [63, 64] (see also [52]), might be a misleading arena for the general story, since it only involves the "trivial" instanton (3.18), and not the more general (and complicated) instantons found in [12, 13] and further clarified in this paper.

Although we have focused in this paper on topological string theory, our results can be applied to any model whose perturbative expansion is described by the holomorphic anomaly equations. This includes systems governed by topological recursion [66], as shown in [67]. In particular, the HAE has proved to be very useful in describing the large $N$ expansion of matrix models [68–70].

The results of this paper can then be used to describe multi-instantons in the large $N$ expansion of matrix models, and therefore in string/gauge theory models with holographic matrix model duals.

Conversely, it would be very interesting to use our results to further our understanding of non-perturbative aspects of the topological recursion, and of large $N$ instantons in matrix models. For example, as a non-trivial consequence of the TS/ST correspondence [20, 21], topological strings on toric CY threefolds and their 4d limits can be described by a new class of convergent matrix models [22, 71–74]. The multi-instantons we have found based on [12, 13] describe the large $N$ instantons of these models, and it would be very interesting to rederive them directly in the matrix model framework, perhaps as some form of eigenvalue tunneling.

Another possible perspective on our results is the following. The genus expansion of topological string theory on a CY manifold can be regarded as the perturbative expansion of a spacetime string field theory, called in [16] the Kodaira–Spencer theory of gravity. Perhaps the multi-instanton amplitudes obtained in this paper can be obtained by doing an expansion around non-trivial saddle-points of this string field theory. A related approach would be to identify our multi-instantons as amplitudes due to (topological) D-branes.

Finally, it would be very interesting to study the trans-series solution of the HAE in the case of compact CYs. The main complications are the presence of additional propagators and the difficulty in producing perturbative data to test the resurgent structure. We expect to report on this problem in the near future [61].

## Acknowledgements

We would like to thank Bertrand Eynard, Amir-Kian Kashani-Poor, Albrecht Klemm, David Sauzin and Ricardo Schiappa for useful comments and discussions. M.M. would like to thank the Physics Department of the École Normale Supérieure (Paris) for hospitality during the completion of this paper. The work of M.M. has been supported in part by the ERC-SyG project "Recursive and Exact New Quantum Theory" (ReNewQuantum), which received funding from the European Research Council (ERC) under the European Union's Horizon 2020 research and innovation program, grant agreement No. 810573. JG is supported by the Startup Funding no. 3207022203A1 and no. 4060692201/011 of the Southeast University.

## A  Useful formulae for local $\mathbb{P}^2$

Here we collect some useful information on topological string theory on local $\mathbb{P}^2$. Most of the formulae below can be found in e.g. [13, 18, 22, 23].

The periods of local $\mathbb{P}^2$ can be obtained by local mirror symmetry [55]. They are built from the formal power series:

$$
\begin{aligned}
\widetilde{\varpi}_1(z) &= \sum_{j \geq 1} 3 \frac{(3j-1)!}{(j!)^3} (-z)^j, \\
\widetilde{\varpi}_2(z) &= \sum_{j \geq 1} \frac{18}{j!} \frac{\Gamma(3j)}{\Gamma(1+j)^2} \left\{ \psi(3j) - \psi(j+1) \right\} (-z)^j,
\end{aligned}
\tag{A.1}
$$

and the flat coordinate at large radius is given by

$$
t = -\log(z) - \widetilde{\varpi}_1(z) = -\log(z) + 6z \, {}_4F_3\left(1, 1, \frac{4}{3}, \frac{5}{3}; 2, 2, 2; -27z\right).
\tag{A.2}
$$

Here, $z$ parametrizes the moduli space of local $\mathbb{P}^2$, and $z = 0$ is the large radius point where $t \to \infty$. The genus zero free energy or prepotential $\mathcal{F}_0(t)$ is defined by

$$\partial_t \mathcal{F}_0(t) = \frac{\omega_2(z)}{6}, \tag{A.3}$$

where

$$\omega_2(z) = \log^2(z) + 2\widetilde{\omega}_1(z)\log(z) + \widetilde{\omega}_2(z). \tag{A.4}$$

It is convenient to define the higher genus free energies in the large radius frame in such a way that the so-called constant map contribution [16] is subtracted. We then have

$$\mathcal{F}_g(t) = \mathcal{O}\left(e^{-t}\right), \tag{A.5}$$

for $t \gg 1$ and $g \geq 2$.

In our parametrization, the conifold point occurs at $z = -1/27$. The discriminant is

$$\Delta = 1 + 27z, \tag{A.6}$$

while the Yukawa coupling is given by

$$C_z = -\frac{1}{3z^3\Delta(z)}. \tag{A.7}$$

The holomorphic function appearing in (2.9) reads

$$f_1(z) = -\frac{1}{12}\log(z^7\Delta), \tag{A.8}$$

and it determines the function $\mathfrak{s}(z)$ in (2.8) through (2.10). The function $\mathfrak{f}(z)$ in (2.11) is given by

$$\mathfrak{f}(z) = \frac{z^4}{4}. \tag{A.9}$$

The flat coordinate at the conifold is given by

$$t_c(z) = \frac{\sqrt{3}}{4\pi}\left(\omega_c(z) - \pi^2\right) \tag{A.10}$$

where

$$\omega_c(z) = \log^2(-z) + 2\log(-z)\widetilde{\omega}_1(z) + \widetilde{\omega}_2(z). \tag{A.11}$$

It has the property that it vanishes at the conifold point:

$$t_c\left(-\frac{1}{27}\right) = 0. \tag{A.12}$$

There is a convenient expression for $t_c$ in terms of a Meijer function:

$$t_c(z) = \frac{3\sqrt{3}}{2\pi}\left(\frac{G_{3,3}^{3,2}\left(-27z \left|\begin{array}{c} \frac{1}{3}, \frac{2}{3}, 1 \\ 0, 0, 0 \end{array}\right.\right)}{2\sqrt{3}\pi} - \frac{4\pi^2}{9}\right). \tag{A.13}$$

This flat coordinate has the following power series expansion near the conifold point,

$$t_c(z) = \Delta + \frac{11}{18}\Delta^2 + \frac{109}{243}\Delta^3 + \cdots, \tag{A.14}$$

where $\Delta$ is the discriminant. The prepotential in the conifold frame is defined by the following small $t_c$ expansion

$$\mathcal{F}_0^c(t_c) = \frac{t_c^2}{6}\log\left(\frac{t_c}{27}\right) - \frac{t_c^2}{4} - \sqrt{3}Vt_c - \frac{t_c^3}{324} + \cdots \tag{A.15}$$

where

$$V = 2\,\mathrm{Im}\,\mathrm{Li}_2\left(\mathrm{e}^{\pi\mathrm{i}/3}\right), \tag{A.16}$$

and we have the relation

$$2\pi\mathrm{Re}(t(z)) = -3\sqrt{3}\frac{\partial\mathcal{F}_0^c}{\partial t_c}. \tag{A.17}$$

The behavior of the higher genus free energies near the conifold point is given by [18, 22]

$$\mathcal{F}_g^c(t_c) = 3^{g-1}\frac{B_{2g}}{2g(2g-2)}t_c^{2-2g} + \mathcal{O}(1), \qquad g \geq 2. \tag{A.18}$$

Therefore, in (2.3) we have $\mathfrak{a} = 1$, $\mathfrak{b} = 3$.

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
