# Peer review of "Exact multi-instantons in topological string theory"

_SciPost Physics_

## Round 2 · Referee Report · Sergiu Vacaru (Referee 1) · 2023-5-24

Strengths

  1. Authors found exact/ closed multi-instanton solutions of equations proposed in [12,13]
  2. Such solutions were generalized to arbitrary multi-instantons.
  3. They stated that the holomorphic limit of these solutions is very simple.

Weaknesses

  1. The results consist of the first step in decoding the full resurgent of the structure of the topological string.
  2. Further tests are necessary, perhaps, in partner work.

Report

The paper meets the journal's acceptance criteria and I consider that it can be published in this journal.

---

## Round 2 · Referee Report · Anonymous (Referee 2) · 2023-6-14

Strengths

  1. Work very clearly presented and with main results generally clearly stated.

  2. General closed-form (exact) results for the multi-instanton sectors of the trans-series as solutions of the holomorphic anomaly equations, providing a much more thorough and complete picture of the free energies/partition function than what existed in the literature.

  3. The operator approach introduced in this work is evidently powerful and bypasses the difficult resursive computations of previous work. The derivations of the computations have a lot of detail, making a very clear distinction between conjectured and derived results.

  4. The examples studied numerically, checking the resurgent nature of the trans-series are in clear support of the derived and conjectured results.

Weaknesses

  1. Derivations quite technical, although this would be expected given the subject of the work, and the level of detail provided in the derivations.

  2. The paper is not completely self contained, as some of the results pertaining to topological string theory and holomorphic anomaly equations are cited but not derived.

Report

This paper presents a very detailed analysis of the full multi-instanton structure of the trans-series solution to the free energies and partition function of topological strings on local Calabi-Yau manifolds with one modulus. The authors re-formulate the holomorphic anomaly equations (HAEs) using an operator formalism. This approach allows them to derive full closed form expressions for all the trans-series sectors, and it is then shown that in the holomorphic limit the multi-instanton sectors become solely dependent on the perturbative free energies. An asymptotic analysis of the large order behavior of these multi-instanton sectors is also given, in the form of their alien derivatives, with conjectures for the Stokes coefficients.

The authors provide a brief review of the main results pertaining the HAEs in Section 2, finishing there with the master equation for the perturbative free energies. This is then followed in Section 3 by the trans-series extension to the HAEs as in previous seminal work (their references [12,13]) and then re-formulate the free energy/partition function trans-series problem in operator formalism, for which they determine previously unknown closed form solutions for all sectors, and analyze some of their properties. Section 4 follows with an analysis of the perturbative free energies in the Borel plane, for the large radius and conifold frames. These confirm their previously derived results and resurgent properties. A summary and discussion of open problems is then given in Section 5.

The paper is very well presented, very detailed and well referenced, with most of the main results highlighted, followed by their derivations. The technical nature of the derivations is expected given the nature of the work. I can only say that I would have liked the paper to be slightly more self-contained in the introduction to topological strings and HAEs, but I do realize that this would amount to a much more lengthy paper, which would likely prove to be less readable in the end. Together with its companion paper on quantum periods, the authors have made a very strong case for their novel operator approach to the multi-instanton sectors of free energies.

It is my opinion that this paper meets of even exceeds the expectations and generally all the acceptance criteria, and it should be published in SciPost. I only have some minor comments/suggestions that could improve the overall readability of the paper.

Requested changes

These are the minor changes/suggestions to take into consideration. Note that some of them are suggestions that would have helped me when reading.

  1. The authors use the expression F_{g}s to mean the different coefficients F_g. I found this confusing: I would suggest replacing "F_{g}s" by "coefficients F_g". This appears on the top of page 6 and in the caption of Figure 2.

  2. Just below equation (3.7), it is mentioned a "trans-series part". But the trans-series is the whole perturbative and non-perturbative sectors, while if I understood correctly the authors here mean the instanton/non-perturbative part of the trans-series.

  3. On page 8, just after eq (3.14), the "polar part of (2.3)" is mentioned. The same thing appear on page 28 just before eq (4.16). What does this polar part mean?

  4. Just before eq (3.24), instead of "using (2.14)" I would suggest "using the relation between propagators (2.14)".

  5. I would suggest adding the final result for the first instanton sector, eq (3.72), when stating the results earlier on. I would first mention (3.72) just below eq (3.42) or around there, so that the result is highlighted instead of buried in the derivation.

  6. Is the vanishing of the r.h.s. of (3.48) completely straightforward? I had some trouble obtaining this. It is very possible that I have missed something, but perhaps adding the main equations one would need to use beyond (2.6) could be helpful?

  7. On page 15, just after eq (3.81) I would suggest "satisfy (2.22)" be replaced by "satisfy master equation (2.22)".

  8. On page 19, after eq (3.124), instead of "behavior (3.18)" I'd suggest "behavior of the boundary condition (3.18)".

  9. Are the particular solutions shown on page 20 special, or are they presented because they are the first representatives of the partition function's sectors? Would be worth mentioning why they were chosen.

  10. On page 23, just before eq (3.148), a "family of trans-series" is mentioned. Why are these trans-series and not just specific sectors of the whole trans-series?

  11. In Figure 1, why wasn't the singularity ell=1 included on the right plot? I believe it would be useful to include it for someone who has little experience on conformal Padé, to see where the different singularities end up appearing.

  12. How difficult would it be to determine the next Stokes coefficients in either (4.12) or (4.22), even numerically?

  13. Would a generalization to more than one modulus be possible?

  • validity: top
  • significance: high
  • originality: high
  • clarity: top
  • formatting: excellent
  • grammar: perfect

Author:  Jie Gu  on 2023-07-09  [id 3787]

(in reply to Report 2 on 2023-06-14)
Category:
remark
answer to question
correction
pointer to related literature
suggestion for further work

  1. This will be fixed.
  2. You are correct. This will be corrected.
  3. The polar part means the part which is a pole. (2.3) is valid for $g\geq 2$, and the polar part simply refers to the order $2g-2$ pole given as the first term on the right hand side.
  4. The suggested rephrasing is better. We will change it.
  5. This is a good suggestion! We will change accordingly.
  6. We added two additional equations with the intermediate results that are needed to verify the vanishing of (3.48) (which has now become (3.49)).
  7. Indeed this is better. We will change it.
  8. We would rather refrain from making the changes as it will lead to a repetition of the phrase "boundary condition" in one short sentence.
  9. These are merely the first few partition functions in instanton/anti-instanton sectors.
  10. In our mind, the concept of trans-series is a bit more general. Sums of trans-series are trans-series, and a component of a trans-series which has the form of "$\sum_i e^{-A_i/g_s}\times$ power series in $g_s$" or "$e^{-A/g_s}\times$ power series in $g_s$" itself is also a trans-series. But if it pleases the referee, we would rephrase "family of trans-series" to "family of trans-series sectors".
  11. Thanks for the suggestion! We will change that.
  12. It is a bit challenging, as we need to first subtract contributions from the leading singularities, and there are multiple singularities at the subleading order. But we are working on it.
  13. The multi-moduli generalization is done in our follow-up paper arXiv:2305.19916.

---

## Round 2 · Referee Report · Anonymous (Referee 3) · 2023-9-8

Strengths

  1. The main results of the paper, that is, exact, closed form multi-instanton solutions for topological strings on local Calabi-Yau threefolds, are novel and strong. To obtain these results, the authors use a new operator formulation for finding transseries solutions to the holomorphic anomaly equations.

  2. The paper is very well written, with a clear exposition. It also provides a nice, concise review of the general idea of finding transseries solutions to the holomorphic anomaly equations.

  3. As a byproduct of the solutions found, the paper clarifies that the multi-instanton solutions can be entirely reconstructed from the perturbative free energies, and thus do not appear to contain new enumerative geometric information.

  4. Beyond the multi-instanton solutions, the paper proposes a conjecture on the form of alien derivatives, which is part of the resurgent structure of the topological string.

  5. The paper provides an explicit example of its main results by studying the topological string on local $\mathbb{P}^2$.

  6. The approach used in the paper could be used more generally, beyond the topological string, as it provides a way to find explicit, closed form, transseries solutions to the holomorphic anomaly equations. Thus, it could be used whenever holomorphic anomaly equations apply; for instance, in enumerative geometric problems governed by the topological recursion of Eynard-Orantin. As such, the paper opens the way for further studies and research directions.

  7. The exact solutions found take a very nice form, which resembles eigenvalue tunneling in matrix models. It thus provides a hint towards further studies on the non-perturbative nature of the topological string in relation with results in other areas such as matrix models.

Weaknesses

  1. The calculations are fairly technical, but this is expected in this research direction.

  2. This relates to Strength #6; the authors could have explored a little further potential applications of their formalism beyond the topological string. But this can certainly be done in further publications, and goes beyond the scope of the present paper.

Report

The main results of the paper are exact, closed form multi-instanton solutions for topological strings on local Calabi-Yau threefolds. The approach used is to find transseries solutions to the holomorphic anomaly equations. This approach is not new, but the closed form solutions found in this paper are certainly novel, and form strong, interesting new results in this research direction. It required using a new operator formulation for the holomorphic anomaly equations, which is the essential new ingredient.

The solutions found take a very nice form, which resembles eigenvalue tunneling in matrix models. This is striking and certainly deserves further studies.

A byproduct of their results is that the multi-instanton solutions can be entirely reconstructed from the perturbative free energies, which is not obvious a priori. In particular, this implies that the multi-instanton solutions do not appear to contain new enumerative geometric information beyond that already contained in the perturbative solutions.

The authors also propose a conjecture on the form of alien derivatives, which is part of the resurgent structure of the topological string.

The example of local $\mathbb{P}^2$ is studied in detail, to make the results of the paper more explicit.

What is also very interesting with the paper is that the proposed operator approach could potentially be used in a more general setting, beyond the topological string. Indeed, the proposed approach relies on the existence of holomorphic anomaly equations satisfied by the perturbative free energies. Thus, it could potentially be used to find multi-instanton solutions in closed form whenever the holomorphic anomaly equations apply. In particular, this includes all the enumerative geometric problems governed by the Eynard-Orantin topological recursion. This opens the way to potential new non-perturbative results in many different contexts.

Overall, there is no doubt in my mind that this paper deserves publication in SciPost. It is a strong paper with new and very interesting results, which form a key step towards obtaining a full understanding of the non-perturbative nature of the topological string. It is also very well written, with a clear and concise exposition.

Requested changes

I only have two very minor comments:

  1. In eq.(3.88) and (3.96), it may be useful to write an explicit bound for the summation on $k$ (I believe it should be $k \geq 1$?)

  2. In eq.(3.139), shouldn't there be an $i$ in front of the summation in the second term on the right-hand-side? (compare with eq.(3.137) )

  • validity: high
  • significance: high
  • originality: high
  • clarity: top
  • formatting: excellent
  • grammar: excellent

Author:  Jie Gu  on 2023-09-11  [id 3973]

(in reply to Report 3 on 2023-09-08)

Dear referee,

thank you very much for you very positive report! We will make changes in response to your two requests. 1. We will make the bounds explicit in eqs. (3.88) and (3.96) (which are now eqs. (3.90) and (3.98)). By the way, the summations start from $k=0$, as shown in the example (3.126) (which is now (3.128)). 2. Yes, you are right. We will include an $i$ in eq. (3.139) (which is now (3.141)).

---

## Editorial Decision

resubmitted